# QKI is a critical pre-mRNA alternative splicing regulator of cardiac myofibrillogenesis and contractile function

Xinyun Chen[1,2,3,8], Ying Liu[3,8], Chen Xu[1,3,8], Lina Ba[3], Zhuo Liu[3], Xiuya Li[1], Jie Huang[1], Ed Simpson [4], Hongyu Gao[4], Dayan Cao[5], Wei Sheng[2,3], Hanping Qi[3], Hongrui Ji[3], Maria Sanderson[3], Chen-Leng Cai[3], Xiaohui Li [5], Lei Yang[3], Jie Na[6], Kenichi Yamamura[7], Yunlong Liu[4], Guoying Huang [2✉], Weinian Shou [3✉] & Ning Sun [1,2✉]

The RNA-binding protein QKI belongs to the hnRNP K-homology domain protein family, a well-known regulator of pre-mRNA alternative splicing and is associated with several neurodevelopmental disorders. *Qki* is found highly expressed in developing and adult hearts. By employing the human embryonic stem cell (hESC) to cardiomyocyte differentiation system and generating QKI-deficient hESCs (hESCs-*QKI^del*) using CRISPR/Cas9 gene editing technology, we analyze the physiological role of QKI in cardiomyocyte differentiation, maturation, and contractile function. hESCs-*QKI^del* largely maintain normal pluripotency and normal differentiation potential for the generation of early cardiogenic progenitors, but they fail to transition into functional cardiomyocytes. In this work, by using a series of transcriptomic, cell and biochemical analyses, and the Qki-deficient mouse model, we demonstrate that QKI is indispensable to cardiac sarcomerogenesis and cardiac function through its regulation of alternative splicing in genes involved in Z-disc formation and contractile physiology, suggesting that *QKI* is associated with the pathogenesis of certain forms of cardiomyopathies.

[1] Department of Physiology and Pathophysiology, School of Basic Medical Sciences, Fudan University, Shanghai, China. [2] Shanghai Key Laboratory of Birth Defects, Children's Hospital of Fudan University, Shanghai, China. [3] Herman B Wells Center for Pediatric Research, Department of Pediatrics, Indiana University School of Medicine, Indianapolis, IN, USA. [4] Department of Bioinformatics, Indiana University School of Medicine, Indianapolis, IN, USA. [5] Institute of Materia Medica and Center of Translational Medicine, College of Pharmacy, Army Medical University, Chongqing, China. [6] Center for Stem Cell Biology and Regenerative Medicine, School of Medicine, Tsinghua University, Beijing, China. [7] Institute of Resource Development and Analysis, Kumanoto University, Kumanoto, Japan. [8]These authors contributed equally: Xinyun Chen, Ying Liu, Chen Xu. ✉email: gyhuang@shmu.edu.cn; wshou@iu.edu; sunning@fudan.edu.cn

The transcriptional and posttranscriptional modifications are critical to mammalian gene expression[1]. RNA splicing is a posttranscriptional process by which introns are removed from the newly transcribed sequences of immature pre-mRNAs and this process is required for generating mature protein-coding mRNAs[2]. In contrast to constitutive splicing, alternative splicing is a dynamic process that is highly regulated upon cellular differentiation or in response to distinct physiological states, resulting in specific exons being either included or excluded in unique combinations to generate multiple mRNA isoforms from a single gene[3]. Genome-wide studies estimated that up to 95% of human genes undergo some level of alternative splicing[4], and these splicing events are associated with almost all normal cellular and physiological functions as well as pathophysiological conditions.

Alternative splicing is regulated by various *cis*-regulatory sequences and *trans*-acting factors[5–8]. RNA-binding proteins (RBPs) can recognize and bind to specific RNA sequences to form ribonucleoprotein complexes and act as regulators of diverse biological functions, such as modifying RNA post-transcriptionally and transporting RNA[9]. Many RBPs play particularly important roles in pre-mRNA splicing[10,11]. In general, upon receiving upstream biological signals, RBPs coordinate alternative splicing events by recognizing specific cis-elements (e.g., enhancers or silencers) to either further promote or inhibit specific splicing events. The most well-known RBP families involved in RNA splicing are the serine/arginine-rich (SR) protein family, with members such as SF2/ASF, SRp20, SRp40, SRp55, and SRp75, and the heterogeneous nuclear ribonucleoprotein (hnRNP) protein family, with members such as PTB, hnRNP A1, hnRNP C, hnRNP D, hnRNP E, hnRNP F/H, hnRNP G, and hnRNP H[12–14]. The heart is the first functional organ to be formed during embryonic development. Interestingly, among ~1148 RBPs in the heart, there are ~390 cardiac-specific RBPs and many of them are present in the developing hearts[15], suggesting that alternative splicing is one of the major regulatory events for cardiogenesis and cardiac physiology. Indeed, it has been shown that mutations of several RBPs and cis-regulatory sequences are associated with cardiomyopathies, muscular atrophy, hypercholesterolemia, and other cardiac disorders[10,16–22].

Previously, several studies have suggested that alternative splicing involved in sarcomerogenesis impacts heart development and function[10]. These prior investigations strongly indicated a critical role for alternative splicing and the associated RBPs in cardiac development and cardiac physiology, as well as the pathogenesis of heart diseases. RBM24 and RBM20 are two of the most well-known RBPs that have been implicated in the pathogenesis of cardiomyopathies[23–25]. The loss of RBM20 function resulted in aberrant splicing events that led to abnormal sarcomerogenesis in embryonic development[24]. Genetic ablation of SRp38 in mice resulted in mis-splicing of triadin, a cardiac protein that regulates calcium release from the sarcoplasmic reticulum during excitation–contraction (E–C) coupling[26]. SF2/ASF cardiomyocyte-specific ablation results in the development of dilated cardiomyopathy and rapid progression of heart failure[27]. RBFox1 acts as a vital regulator for the conserved splicing process of transcription factor MEF2 family members and is involved in the heart failure progression[28].

One particular member of the STAR (signal transduction and activation of RNA) gene family, known as Quaking (QKI), belongs to the hnRNP K-homology (KH)-domain family of proteins, and it is a sequence-specific RBP that is enriched in the heart and central nervous system[29–32]. QKI contains an RNA-binding motif (KH domain), which is flanked by QUA1 and QUA2 domains. The QUA domain is involved in forming homo- or heterodimers and is required for RNA binding[33–36]. In addition to these functional domains, there is a tyrosin cluster located within the proline-rich PXXP motif that can be phosphorylated by Src kinases[37], suggesting that QKI can be regulated by intracellular signaling. At least three major alternatively spliced mRNA isoforms are generated from the QKI gene, QKI−5, QKI-6, and QKI-7, in which exons 1–6 encode identical structures in these isoforms but differ in their C-terminal end encoded by exons 7 and 8[38]. QKI-5 has a nuclear localization signal (NLS)[39,40] and has been shown to play a major function in pre-mRNA splicing regulators[41–46]. QKI-6 and QKI-7 lack NLS and apparently have different biological functions[38,40,47,48]. They seem to play more important roles in regulating mRNA stability and other post-transcriptional mRNA processing and intracellular transportation[49–52]. These functional differences reflect the complexity of QKI biological functions. Based on the early embryonic lethal phenotype of Qki-deficient mice, Qki is considered essential for early embryonic development[53]. Interestingly, a spontaneous mutant mouse line, known as the Qk[v] mouse line, in which a 1 Mb promoter/enhancer deletion upstream of the Qki transcription start site resulted in deficient myelination in the central nervous system[34,54]. Over the past decades, the use of these Qki mutant animal models has suggested important functions in neural progenitors, myelin formation, smooth muscle differentiation, and monocyte to macrophage differentiation[45,55,56]. Despite the evidence of Qki expression in developing hearts, the role of QKI in regulating normal cardiogenesis and cardiac physiology has not been carefully studied.

In this work, we analyze the biological function of QKI-mediated post-transcriptional regulation in cardiogenesis and cardiac physiology. By taking advantage of a defined in vitro system for differentiating human embryonic stem cells (hESCs) into cardiomyocytes and the Qkiβ[Geo] mouse model, we are able to reveal that QKI is a critical pre-mRNA alternative splicing regulator in cardiomyocytes and is essential for myofibrillogenesis and contractile physiology during cardiomyocyte differentiation and maturation. Our finding shows that QKI is indispensable to normal cardiogenesis and cardiac function.

## Results

**The expression of *QKI* in cardiomyocytes and the generation of hESC-QKI^del.** QKI expression was previously found in the hearts[29]. To determine the temporal expression pattern of *QKI* during hESC-to-cardiomyocyte differentiation (Supplementary Fig. 1A), we analyzed *QKI-5, -6,* and *-7* mRNA expression levels in undifferentiated hESCs and differentiating cells at Day-1, -3, -6, -8, and -10 by using quantitative reverse transcription PCR (qRT-PCR) (Fig. 1A). In this in vitro differentiation system, cells at Day-1 were considered to be early nascent mesodermal cells with high levels of expression of *T* (*Brachyury*); cells at Day-3 were considered to be cardiogenic mesodermal cells with a specific upregulation of *MESP1*; cells at Day-6 were considered to be cardiac progenitor cells with upregulation of *ISL1*; and cells at Day-8 to -10 were considered to be well-committed differentiating cardiomyocytes positive for a series of cardiomyocyte markers, such as *NKX2-5* and *TNNT2*. Using these markers as internal references (Fig. 1B), we were able to demonstrate that *QKI-5* transcripts were present in undifferentiated hESCs and there was a quick downregulation at the early differentiation stage. It maintained at a steady lower expression level until differentiation Day-6 to Day-10 (Fig. 1A), at which point cells reached the transition stage from cardiogenic progenitors to early differentiated cardiomyocytes. Comparing to *QKI-5* expression, *QKI-6* and *QKI-7* expression levels were at a much lower level throughout, while *their* expression was elevated in the later stage of differentiation (Fig. 1A). Western blot analysis further

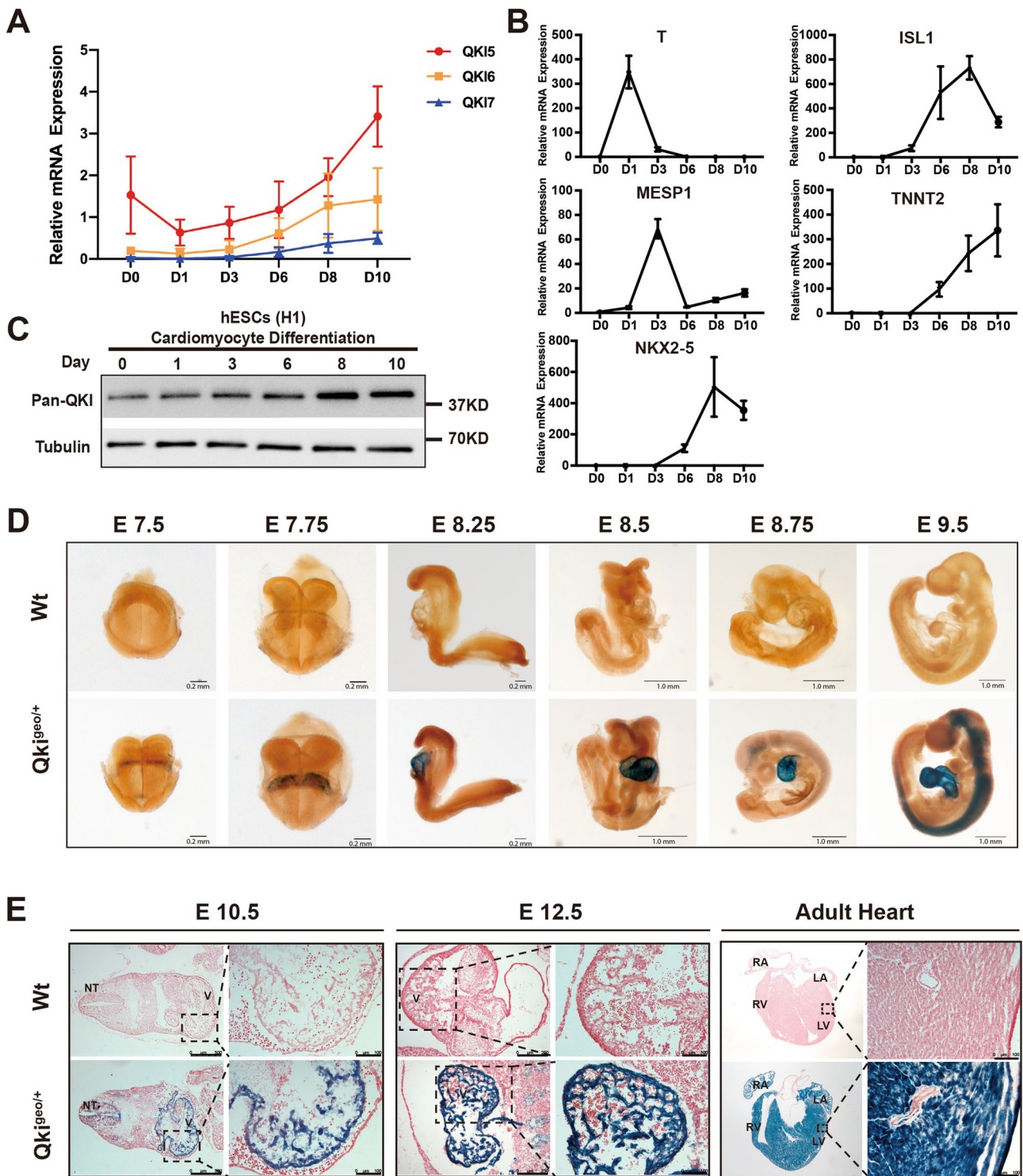

supported this observation that QKI-5 is the main isoform in hESCs and during hESC-to-cardiomyocyte differentiation (Fig. 1C and Supplementary Fig. 1B–E). This finding indicated that *QKI* was likely to have a specific function in the early cardiogenic process of transitioning cardiac progenitors to early cardiomyocytes. To confirm this observation in vivo, we analyzed the *Qki* spatiotemporal expression pattern in developing mouse embryos from E7.5 to E9.5 using the previously generated *Qki^{βGeo}* mouse model (Fig. 1D, E); in this mouse line, the *βGeo*-reporter expression cassette is under the control of *Qki* promoter[53], thus representing *Qki* expression. Based on the X-Gal

(5-bromo-4-chloro-3-indolyl-β-D-galactopyranoside) staining, we were able to visualize reporter expression in the developing hearts from the E7.5 cardiac crescent to the four-chamber heart and the expression was persistent in both the atrial and ventricular myocardia (Fig. 1D, E), confirming that *QKI* was likely critical for early cardiogenic events. Cardiac *Qki* expression persisted in the postnatal hearts (Fig. 1E), further indicating a potentially important role for *QKI* in maintaining normal adult cardiac function.

*QKI*-deficient mutant hESC lines (hESC-*QKI^{del}*) were generated by using CRISPR/Cas9 gene-editing technology. The guide

**Fig. 1 QKI expression pattern in cardiac differentiation. A** qRT-PCR analysis of QKIs-5, -6, and -7 mRNA expression during differentiation of hESCs into cardiomyocytes. Data are shown as the mean ± SEM from five independent experiments. **B** Expression of *T* (Brachyury), *Mesp1*, *Nkx2.5*, *ISL1*, and *TNNT2* was used to intrinsically mark various differentiation stages between hESCs and cardiomyocytes. Data were normalized to Ribosomal Protein L7 (RPL7). Data are shown as the mean ± SEM from three independent experiments. **C** Western blotting using anti-pan-QKI antibody to determine the protein expression level of QKI during hESC-CM differentiation at Day-0, -1, -3, -6, -8, and -10. Based on the additional western blottings showing the relative protein levels of QKI-5, QKI-6, and -7 using isoform specific antibodies (Supplementary Fig. 1), QKI-5 is the dominant isoform from hESCs to differentiated cardiomyocytes. The experiements are independently repeated five times with multiple different sets of collected cell culture samples to ensure the reproducibility of the result. **D** Whole-mount X-Gal staining of *Qki^{geo/+}* mouse embryos shows the spatiotemporal expression pattern of *Qki* in mouse early embryos. Scale bar: 0.2 μm and 1.0 mm, respectively. Total of six embryos of each indicated stage of wild-type and *Qki^{βGeo/+}* embryos from four to five independent timed-mating litters are collected and used in the experiments. **E** Embryos at E10.5, E12.5, and adult hearts were sectioned and stained with X-Gal and fast red. Blue signals indicate the positive expression of *Qki*. Scale bar: 100 μm. Total of six embryos of each indicated stage of wild-type and *Qki^{βGeo/+}* embryos from five independent timed-mating litters and five pairs of adult heart samples (2-month-old) are collected and used in the experiments.

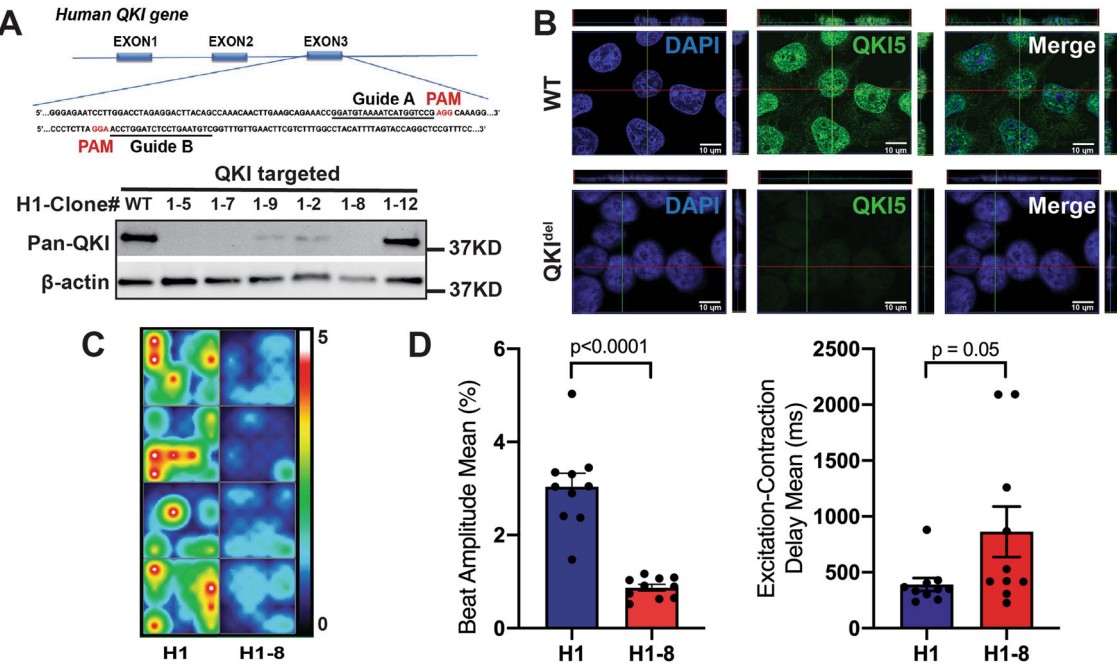

**Fig. 2 The generation and characterization of hESCs-*QKI^{del}*. A** Schematic diagram of the exon 3 targeting site for designing guide RNAs using the CRISPR/Cas9 strategy and a representative Western blot screening for positively targeted clones, which is confirmed by additional two repeats of western blottings and followed by sequencing confirmation. **B** Representative confocal images of immunofluorescence staining confirming the genetic ablation of *QKI-5*. No detectable positive QKI-5 expression was found in mutant H1-8 cells, whereas there was strong nuclear QKI-5 expression (green fluorescence signal) in control H1 cells. Three independent sets of cells are used in the analysis and the finding is consistently confirmed. **C** Representative heatmap of Maestro-MEA assay for contractile function comparing Day-30 cardiomyocytes derived from control hESCs (H1) and hESCs-*QKI^{del}* (H1-8). **D** Comparison of the relative contractility and the excitation–contraction delay between Day-30 cardiomyocytes derived from control hESCs (H1) and hESCs-*QKI^{del}* (H1-8). Significantly reduced contractility and prolonged E–C delay in cardiomyocytes derived from hESCs-*QKI^{del}* when compared to cardiomyocytes derive from control hESCs. Data are shown as the mean ± SEM and statistical significance was determined by a two-tailed Student's *t*-test.

RNAs were designed to target the third exon of *QKI* (Fig. 2A, upper panel). Initial loss-of-expression clones were screened by Western blot with an anti-pan-QKI antibody (Fig. 2A, lower panel, and Supplementary Fig. 2A). Positive clones with complete loss-of-expression were subjected to sequencing confirmation, followed by careful confocal imaging of immunofluorescence staining with an anti-QKI5 antibody (Fig. 2B). The complete loss of the specific nuclear staining validated the successful generation of hESC-*QKI^{del}* mutant lines. We were able to derive a total of 6 hESC-*QKI^{del}* mutant lines from three established hESC lines, H1, H7, and H9. As all hESC-*QKI^{del}* clones showed similar phenotypes, hESC-*QKI^{del}* congenic lines derived from H1, namely, clone H1-7 and clone H1-8 (Supplementary Fig. 2B), were used in the following analyses.

**Altered contractile function of cardiomyocyte sheets derived from hESCs-*QKI^{del}*.** As *QKI* is expressed in undifferentiated hESCs, we first analyzed the pluripotency and self-renewal capacity of hESCs-*QKI^{del}* by examining the expression of alkaline phosphatase and the following pluripotent markers: Oct4, Sox2, Nanog, c-Myc, Klf4, SSEA-4, and Tra-1-60. As shown in Supplementary Fig. 3A–C, hESCs-*QKI^{del}* had normal expression of these pluripotent markers, indicating that *QKI* was not required for maintaining the pluripotency of hESCs. To further confirm this conclusion, we injected *QKI^{def}*-hESCs into nonobese diabetic/severe combined immunodeficiency mice to test whether they could form teratomas (containing all three germ layers). Our data confirmed the formation of well-formed teratomas containing the three embryonic germ layers derived from

hESCs-QKI[del] (Supplementary Fig. 3D, E). In conclusion, hESCs-QKI[del] were normal in the maintenance of pluripotency and the differentiation potentials to the three germ layers.

To test whether *QKI* was specifically involved in cardiomyocyte differentiation and function, we analyzed the cardiomyocyte differentiation efficiency of hESCs-QKI[del] (e.g., H1-7 and H1-8) in comparison to congenic control hESCs (e.g., H1 cells). Consistent with the different hESC-QKI[del] clones tested, H1-7 and H1-8 were able to differentiate into monolayer cardiomyocyte sheets but with a dramatically weak and asynchronous spontaneous-beating activity at Day-15; H1-7- and H1-8-differentiating cells further progressed to near none-beating cardiomyocyte sheets at Day-30, which was distinct from the strong and synchronized spontaneous-beating monolayer of cardiomyocyte sheets derived from control hESCs at both Day-15 and Day-30 (Supplementary Movies 1–6). Quantification of contractility by using Maestro-MEA assay demonstrated dramatically reduced contractility and increased E–C delay (Fig. 2C, D). These observations suggested a significant defect in cardiomyocyte differentiation, maturation, or physiological function in differentiated cardiomyocytes derived from hESCs-QKI[del] (hCMs-QKI[del]). When we used flow cytometry to analyze the percentage of TNNT2-positive cells at Day-15, we found that there was an approximately 20% reduction of TNNT2-positive cells in H1-7 and H1-8 cardiomyocyte sheets when compared to control H1 cardiomyocyte sheets (Supplementary Fig. 4), suggesting that some degrees of cardiomyocyte differentiation were affected in hESCs-QKI[del].

**Single-cell transcriptomic analysis of hCMs-QKI[del]**. Despite the reduction in TNNT2-positive cells, it was difficult to interpret the dramatically reduced contractile activity in well-formed cardiomyocyte sheets derived from hESCs-QKI[del] based solely on the reduced number of TNNT2-positive cells. To better determine whether specific cardiac developmental or cardiomyocyte differentiation events were affected in hESCs-QKI[del], we performed single-cell transcriptome (single-cell RNA sequencing (scRNA-seq)) analysis and compared H1 and H1-8 cells at Day-6 and -15 of differentiation. A total of over 21,920 cells were captured and 21,443 cells were retained for further analysis after quality control (QC) steps. We were able to analyze the expression of over 6500 genes from these cells. Based on the unsupervised computation of the preserved similarity of genome-wide expression, these cells were clustered into 19 groups in the integration analysis of data from Day-6 and -15 samples, and the clustering was visualized by color coding in Uniform Manifold Approximation and Projection (UMAP) plots (Fig. 3A). These clusters could be classified into two large groups, cardiogenic clusters that accounted for over 85% of total cells, and non-cardiogenic clusters (Fig. 3A). When comparing the overall distribution of differentiated cells derived from hESCs-QKI[del] and normal control hESCs at Day-6 and -15 (Fig. 3B), we did not observe an obvious defect in terms of different cell lineages, except for some minor clusters that were towards non-cardiomyocyte lineages (e.g, cluster 17, a small cluster of cells towards neural progenitors), suggesting that the overall lineage specification and determination towards cardiomyocytes were largely preserved in hESCs-QKI[del]. The representative violin plots of several key cardiogenic transcription factors (e.g., *ISL1*, *NKX2.5*, and *TBX5*) and cardiomyocyte functional markers (e.g., *RYR2*, *PLN*, and *TNNT2*) demonstrated the dynamic distribution of expression in all clusters at Day-6 (Fig. 3C) and Day-15 (Fig. 3D). Based on these key molecular markers and the additional signature functional genes (Supplementary Fig. 5A), we annotated these cardiogenic clusters into 6 differentiation states (Fig. 3E), which include the cardioprogenitors, early differentiating cardiomyocytes, high proliferative cardiomyocytes, differentiated cardiomyocytes, endothelial/blood lineage cells, and epicardial lineage cells.

To better understand the differentiation and maturation process, additional analysis of scRNA-seq data was performed by using Seurat 3[57,58] and Monocle algorithm (Monocle 3)[59] to determine the trajectories in pseudotime during hESC-to-cardiomyocyte differentiation. As shown in Fig. 3E, pseudotime ordering of cardiogenic cells derived from hESCs-QKI[del] and normal control hESCs from early cardioprogenitors to differentiated cardiomyocytes using cluster 12 as the root cells. Under our hESC-to-cardiomyocyte differentiation protocol, early cardioprogenitors branched out two major populations of cardiomyocytes, higher proliferative cardiomyocytes, which had enriched expression of genes marking cells in the active proliferative state (Supplementary Fig. 5A), and differentiated cardiomyocytes with lower proliferative activity. Apparently, the differentiated cardiomyocytes were derived from two routes; the first route was from cardioprogenitors (cluster 12) to higher proliferating cardiomyocytes (clusters 6 and 8) and then to lower proliferative cardiomyocytes (clusters 0 and 4) (Fig. 3E), which occurred mainly in early differentiation stage (Day-6, Fig. 3F); the second route was from cardioprogenitors (cluster 12) to early differentiating cardiomyocytes (cluster 2) and differentiated cardiomyocytes (clusters 0 and 4) (Fig. 3E, G). Importantly, all states of cells towards cardiomyocytes differentiated from hESCs-QKI[del] and normal control hESCs were largely superposable in Monocle trajectory analysis, confirming overall differentiation trajectory were consistent between WT control hESCs and hESCs-QKI[del] under our differentiation protocol. The number of cardioprogenitors and early differentiating cardiomyocytes were similar in H1-hESC differentiation group and hESCs-QKI[del] group (10.9% in H1 vs. 9.71% in H1-8), suggesting that the cardiomyocyte lineage specification, determination, and early differentiation were largely preserved in hESCs-QKI[del]. However, it was important to note that the differentiated cardiomyocytes (clusters 0 and 4) accounted for 53.1% of total cells in the normal control hESC differentiation group and 43.3% in hESCs-QKI[del] group at Day-15, a roughly 18.5% reduction. This finding was similar to the significant reduction of TNNT2-positive cardiomyocytes in hESCs-QKI[del] measured by fluorescence-activated cell sorting analysis (Supplementary Fig. 4). Interestingly, differentiating cardiomyocytes (cluster 2) at Day-15 accounted for 2.14% of total cells in the normal control group; however, it accounted for 4.97% of total cells in hESCs-QKI[del] group, a dramatic 132.0% increase. This correlation further suggested that the major defect in hESCs-QKI[del] was at the transition stage from early differentiating cardiomyocytes to functionally more mature cardiomyocytes.

Based on the signature gene expression profiles, in all cardiogenic and early cardiomyocyte clusters at D-6, we did not find a major alteration in gene expression between the hCMs-QKI[del] and the control cells (Supplementary Fig. 5B). To confirm this observation, we used qRT-PCR to analyze the temporal expression patterns of several key mesodermal and cardiogenic lineage markers from Day-0 to Day-10: *MESP1*, *Brachyura (T)*, *ISL1*, *NKX2-5*, *GATA4*, *MYOCD*, *TNNT2*, and *ACTN2*. As shown in Supplementary Fig. 6A, except for the downregulation of *ACTN2* at Day-10, the overall expression pattern of these major lineage markers was largely unchanged during the early differentiation process. These data confirmed that the majority of hESCs-QKI[del] had normal early cardiomyocyte specification and determination. In the late cardiomyocyte clusters of Day-15 (clusters 0 and 4), we found that the altered expression of genes was most significant to contractile function (Fig. 3H and Supplementary Fig. 6B). Among them, *ACTN2* (α-actinin),

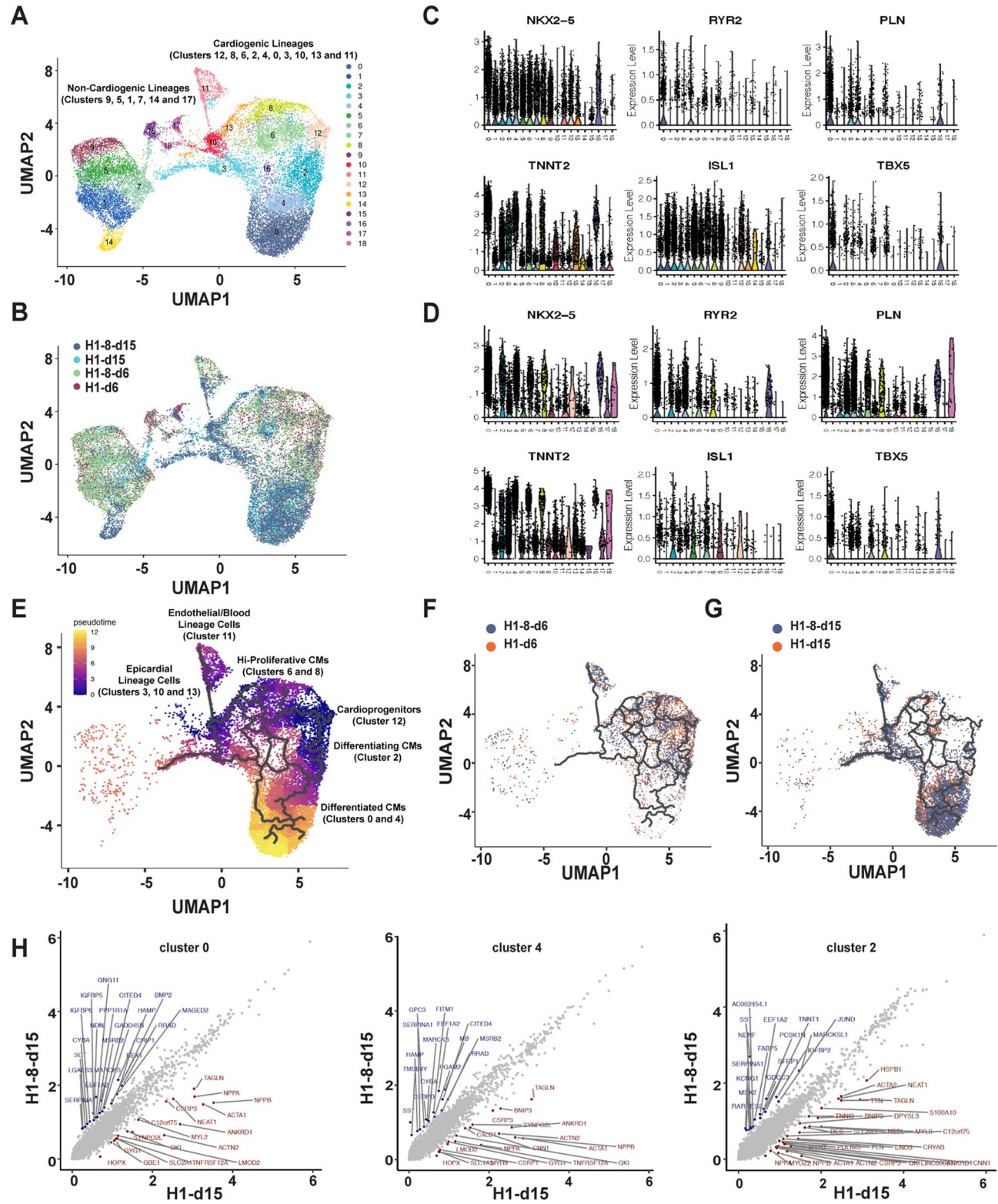

TNNI3 (troponin I3), MYL2 (myosin light chain 2), MYH7 (β-myosin heavy chain), ACTA1 (actin α-cardiac muscle 1), ANKRD1 (ankyrin repeat domain 1), and TAGLN (transgelin) were significantly downregulated in the hCMs-QKI$^{del}$, suggesting that the diminished contractile function in cardiomyocyte sheets derived from hESCs-QKI$^{del}$ was largely due to the altered regulation of sarcomere genes. Interestingly, the most significantly altered genes in the early differentiating cardiomyocytes

(cluster 2) included genes involved in post-translational protein phosphorylation and cell cycle regulation (Fig. 3H and Supplementary Fig. 6B), suggesting a defect in the intracellular signaling. The downregulation of ACTN2 was confirmed in Day-15 hCMs-QKI$^{del}$ (Supplementary Fig. 6C). In addition, the ratio of MYH7/MYH6 in hCMs-QKI$^{del}$ at Day-15 was significantly lower compared to control hCM (Supplementary Fig. 6C), indicating a defect in the process of functional maturation of cardiomyocyte.

**Fig. 3 Single-cell transcriptome analysis of hESCs and hESC-*QKI^del* differentiated cells. A** An UMAP plot shows unsupervised clustering of 19 groups from integrated cell data of differentiated cells at differentiation Day-6 and Day-15. Each cluster is color-coded. The major clusters are to cardiogenic lineages, along with minor clusters of non-cardiogenic lineages. **B** In the comparison of the distribution of differentiated cells, the overlay of differentiated cells integrated with control H1 and mutant H1-8 cells at Day-6 and Day-15 shows a largely superposable distribution. **C**, **D** Representative violin plots show the expression of several key cardiac genes, such as *NKX2-5*, *ISL1*, and *TBX5*, *RYR2*, *PLN*, and *TNNT2* in each cluster at Day-6 (**C**) and Day-15 (**D**), demonstrating dynamic changes of gene expression profile during differentiation. **E** Monocle 3 analysis of trajectory in pseudotime of cardiogenic lineage cells shows six major cell states from early cardioprogenitors towards differentiated cells, which includes cardioprogenitors, high proliferative cardiomyocytes (hi-Proliferative CMs), differentiating cardiomyocytes (Differentiating CMs), differentiated cardiomyocytes (Differentiated CMs), Endothelial/blood lineage cells, and epicardial lineage cells. Cluster 12 was used as root cells for generating the trajectory of cardiogenic differentiation. **F** At Day-6, cardioprogenitors, high proliferative cardiomyocytes, and differentiating cardiomyocytes are the major cell types. **G** Differentiated cardiomyocytes are the major cardiomyocytes at Day-15, accounted for 53.1% and 43.3% in control hESCs (H1)- and hESCs-*QKI^del* (H1-8)-derived cells, respectively. **H** Representative scatter plots of average gene expression profiles of differentiated cardiomyocytes (clusters 0 and 4) and differentiating cardiomyocytes (cluster 2) of control H1 versus H1-8 at Day-15. See Supplementary Fig. 6B for respective Reactome Pathway Analysis of clusters 0, 4, and 2.

---

These data further confirmed that the role of *QKI* in cardiomyocyte differentiation was at the transition stage bridging early cardiomyocyte to functional cardiomyocytes, which was consistent with the overall *Qki* spatiotemporal expression pattern in mouse early embryos as well as in the hESC-to-cardiomyocyte differentiation process (Fig. 1).

**Bulk transcriptomic analysis of hCMs-*QKI^del*.** To further confirm this observation, a bulk RNA-seq analysis was performed, which provided a more in-depth transcriptome comparison between Day-15 differentiated cells of hESCs-*QKI^del* (H1-8) and control hESCs (H1). As demonstrated by the volcano plot in Fig. 4A, based on the false discovery rate (FDR) value of <0.05 and the fold change of 1.5, 556 genes were upregulated, and 661 genes were downregulated. Based on Gene Ontology (GO) analysis (Fig. 4B), muscle contraction, muscle tissue development, and animal organ morphogenesis were among the most affected biological processes (BP). Interestingly, despite the RNA-seq being performed using the differentiated cardiomyocytes, other top affect BP included the regulation of *trans*-synaptic signaling and the regulation of blood vessel diameter; the former was likely consistent with the notion that *Qki* was highly expressed in the developing and adult nervous system (Fig. 1)[60,61], and the latter was likely consistent with the vascular developmental defect in *Qki*-deficient embryos[53]. The GO analysis also indicated the most significantly affected molecular functions (MFs) and cellular components (CCs), which were consistent with the finding in altered BP. Furthermore, KEGG (Kyoto Encyclopedia of Genes and Genomes) pathway analysis indicated that pathways involved in cardiomyopathies, regulation of actin cytoskeleton, cardiac muscle contraction and adhesion, mitogen-activated protein kinase signaling, Rap1 signaling, and adrenergic signaling were the most significantly altered biological pathways (Fig. 4C). Figure 4D represents the heatmap of differentially expressed genes within these affected BPs and pathways. This finding was confirmed by additional investigation using Ingenuity Pathway Analysis (Fig. 4E, F). Collectively, our data strongly suggested that *QKI* was most likely a key regulator of cardiomyocyte myofibrillogenesis and contractile function.

**QKI-mediated alternative splicing in cardiac myogenesis.** The major MF of QKI is its regulation of alternative splicing. The data from transcriptome profiling as discussed above mostly reflected the altered cellular function and the pathogenetic pathways associated with altered function in hCMs-*QKI^del*. Based on the bulk RNA-seq data, we analyzed differential alternative splicing events in hCMs-*QKI^del* using rMATs software[62]. Based on the FDR value (<0.05) and the δPSI value (> ±0.3), we found that all 5 alternative splicing types were present in hCMs-*QKI^del*, including

skipped exon (SE), intron retention, mutually exclusive exons (MXEs), alternative 5′-donor sites, and alternative 3′-acceptor sites (Fig. 5A). Among them, SE and MXE were two major alternative splicing types in hCM-*QKI^del*, which accounted for 93% of total AS events (among 461 total alternative splicing events, there were 363 SE events and 65 MXE events) from a total of 299 genes (Fig. 5A). Based on the RBP motif analysis, the majority of these alternatively spliced events contained the QKI RNA-binding consensus sites (Supplementary Fig. 7)[63], confirming that the majority of differential alternative splicing events in hCM-*QKI^del* were relevant to QKI-mediated processes. Interestingly, GO analyses of these altered alternative splicing events suggested that the most significantly affected BPs in hCM-*QKI^del* were the actomyosin structure organization and the regulation of actin filament organization (Fig. 5B), in which genes involved in cardiac myofibrillogenesis and cardiomyocyte contractile function, such as *ACTN2*, *ABLIM1*, *TTN*, *NEBL*, *RYR2*, and *CACNA1G*, were affected, as confirmed by RT-PCR (Fig. 6A), suggesting that the primary defect in hCM-*QKI^del* was the dysregulated formation of contractile apparatus and its associated physiological function, which was further confirmed by the most significantly affected events in MFs and CCs, as well as the molecular pathways demonstrated by KEGG pathway analysis, not only indicating that myofibrillogenesis was the major affected process but also its potential link to cardiomyopathies (Fig. 5B, C). Figure 6B shows the representative schematic diagrams of altered SE events in *ACTN2*, *ABLIM1*, *NEBL*, *RYR2*, *CACNA1C*, and *PDLIM5*, which were derived from rMATs analysis and sashimi plots shown in Supplementary Fig. 8. As *QKI* is known for its sequence-specific RNA binding, we used RNA immuno-precipitation (RIP)-quantitative PCR (qPCR) to confirm that several major SE events were QKI-mediated. As shown in Fig. 6C, we were able to validate that QKI was able to bind to ACTN2, CAMK2D, PDLIM5, NEBL, ABLIM1, RYR2, and CACNA1C. Collectively, these data confirmed the role of QKI in regulating alternative splicing in genes critically involved in sarcomere formation and contractile function.

**QKI-mediated alternative splicing in the *ACTN2* gene and altered myofibrillogenesis in hCMs-*QKI^del*.** When we compared the differentially expressed genes with genes that underwent differential alternative splicing in hCM-*QKI^del*, we found that eight genes were associated with both groups (Supplementary Fig. 9A). Among them, *ACTN2* was at the top in both the downregulated gene list (reduced over 29-fold) and the differential alternative splicing list. In hCMs-*QKI^del*, *ACTN2^SE-8* was generated, which had an abnormally skipped exon 8a or exon 8b (FDR < 0.0000001). *QKI*-binding consensus sequences were mapped in *ACTN2* exon 7, and in introns flanking exons 8a and 8b in *ACTN2* isoform 1 and isoform 2 (Fig. 6D). Interestingly, the

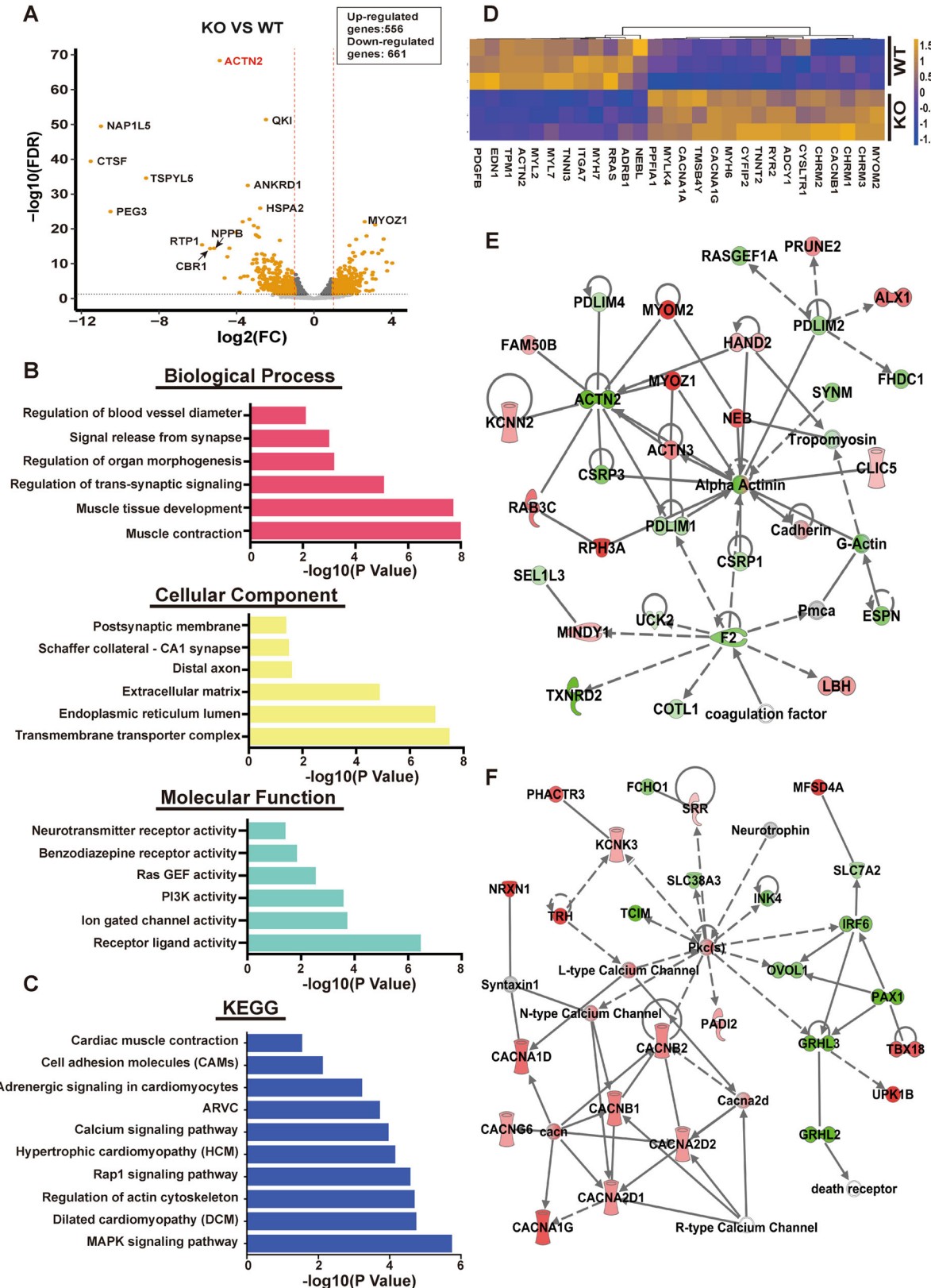

analysis of the mRNA sequence of *ACTN2^SE-8^* revealed that a premature stop codon appeared in a position located in exon 9 (Fig. 6D and Supplementary Fig. 9B). To confirm whether this premature stop codon caused an activation of nonsense-mediated decay (NMD), which subsequently led to the dramatic down-regulation of *ACTN2* expression, we applied NMDI14 (Millipore

sigma; SML1538) to hCM-*QKI^del^*, a specific inhibitor disrupting the interaction of UPF1 and SMG7 in the NMD pathway[64]. We found that *ACTN2^SE-8^* mRNA levels were significantly enriched in NMDI14-treated cells (Fig. 6D, right panel), indicating that this rapid degradation of *ACTN2^SE-8^* mRNA in hCM-*QKI^del^* was in fact due to the activation of the NMD pathway. Collectively,

**Fig. 4 Bulk transcriptome analysis of hESCs and hESC-*QKI^del* differentiated cardiomyocytes at Day-15. A** A representative volcano plot demonstrates differentially expressed genes between H1 differentiated cardiomyocytes and mutant H1-8 differentiated cardiomyocytes at Day-15. Based on the FDR value (<0.05) and fold change (FC), over 1556 genes were upregulated, and 661 genes were downregulated in H1-8 differentiated cardiomyocytes. **B**, **C** Representative enrichment analysis of differentially expressed genes using Gene Ontology (GO) (**B**) and KEGG pathways (**C**) databases. The top 6 ranked GO terms altered in the categories of Biological Process, Cellular Component, and Molecular Function, and the top ten ranked KEGG terms are represented in the bar graph. GO enrichment and KEGG pathway enrichment analysis of differentially expressed genes were performed respectively using R programming based on the hypergeometric distribution. **D** Representative heatmap shows differentially expressed genes relevant to cardiac muscle contraction and the calcium signaling pathway. **E**, **F** Representative IPA core analysis of key terms associated with cardiovascular diseases and the interaction network. The upregulated and downregulated genes are marked in red and green, respectively.

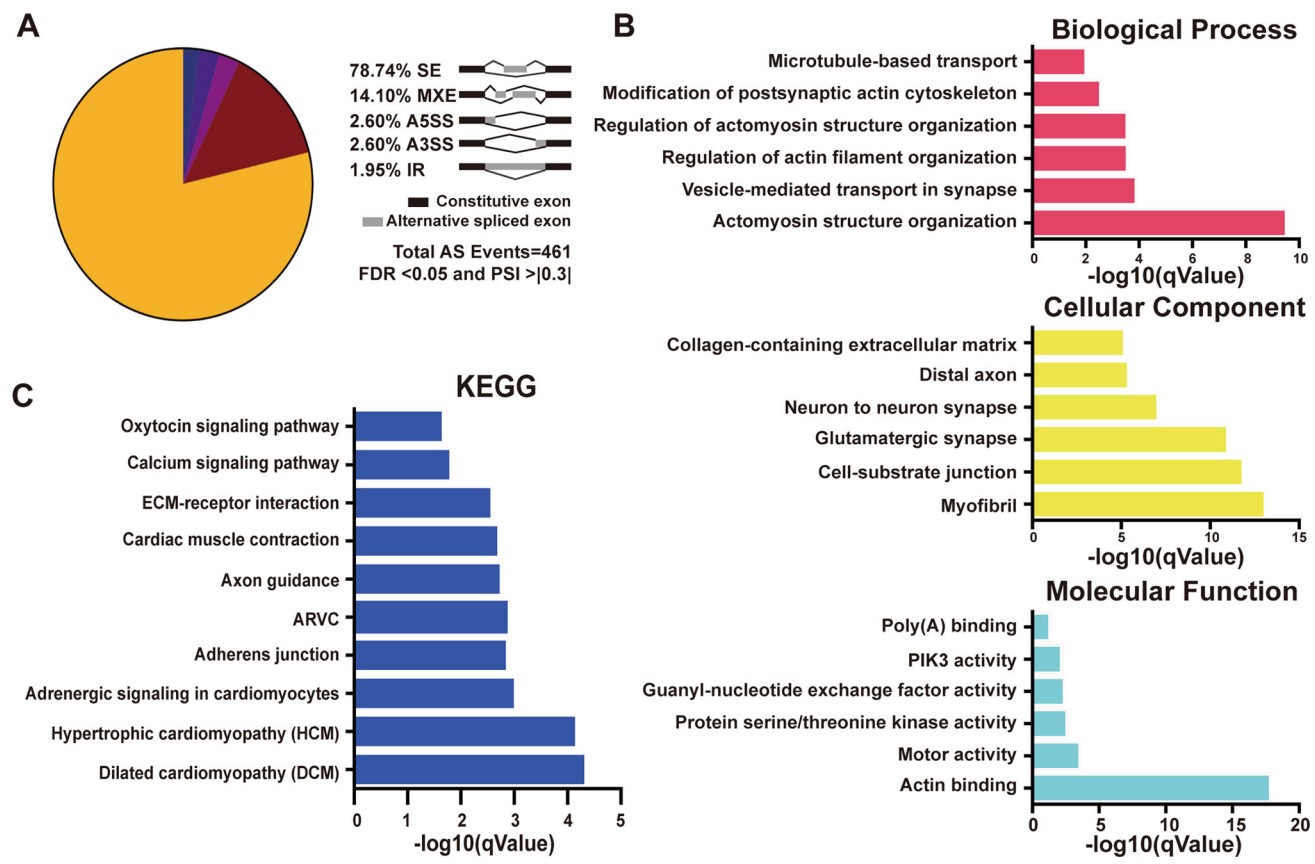

**Fig. 5 Analysis of differential alternative splicing events in hESC and hESC-*QKI^del* differentiated cardiomyocytes at Day-15. A** Schematic diagram shows the overall distribution of alternative splicing events in hESC-*QKI^del* differentiated cardiomyocytes. **B** Enrichment analysis of differentially spliced genes using Gene Ontology (GO) and **C** KEGG pathway databases. GO enrichment and KEGG pathway enrichment analysis of differentially expressed genes were performed, respectively, using R programming based on the hypergeometric distribution.

these findings not only confirmed that *ACTN2* was a primary target of *QKI* but also explained the cause of the dramatic downregulation of *ACTN2* mRNA expression level in hCM-*QKI^del*.

To further investigate myofibrillogenesis and the cause of diminished contractile function in hCMs-*QKI^del*, we analyzed myofibril structures in hCMs-*QKI^del* and compared them to those in normal control hCMs by performing dual-immunofluorescence staining for TNNT2 and ACTN2. As shown in Fig. 7A, well-organized striated myofibril structures were found in control hCMs, which was in great contrast to the largely disorganized myofibrils lacking striated structure in hCMs-*QKI^del*. The immunoreactivity to anti-ACTN2 was severely reduced in hCMs-*QKI^del*, consistent with the downregulation of *ACTN2* mRNA level. The disruption of the myofibril structure in hCMs-*QKI^del* was further confirmed by other sets of dual-immunofluorescence staining experiments, using antibodies against NEBL/TNNT2 (Fig. 7B) and TTN/TNNT2 (Supplementary Fig. 10). To further confirm this result, transmission

electron microscopy (TEM) analysis was performed. As expected, thinned and disorganized sarcomeres lacking Z-disc structures were found in hCMs-*QKI^del*, which was contrasted with the well-formed sarcomeres seen in control hCMs (Fig. 7C). Furthermore, in addition to ACTN2, Western blot analysis demonstrated significantly reduced protein levels of multiple sarcomeric proteins, including the myosin heavy chains (i.e., positive for MF20 antibody), TNNT2, MYOZ2, and TNNI3 (Fig. 7D), further confirming a defect in myofibrillogenesis in the hCMs-*QKI^del*.

**QKI-5-mediated RNA-splicing activity is key to regulate cardiac myofibrillogenesis.** QKIs have three major splicing isoforms, known as QKI-5 (QKI-203), QKI-6 (QKI-207), and QKI-7 (QKI-201), which differ in the last two coding exons 7 and 8. All QKI isoforms were present in heart tissue, whereas QKI5 was the highest-expressed isoform. QKI-5 contains an NLS and was exclusively localized in cardiomyocyte nuclei, which was

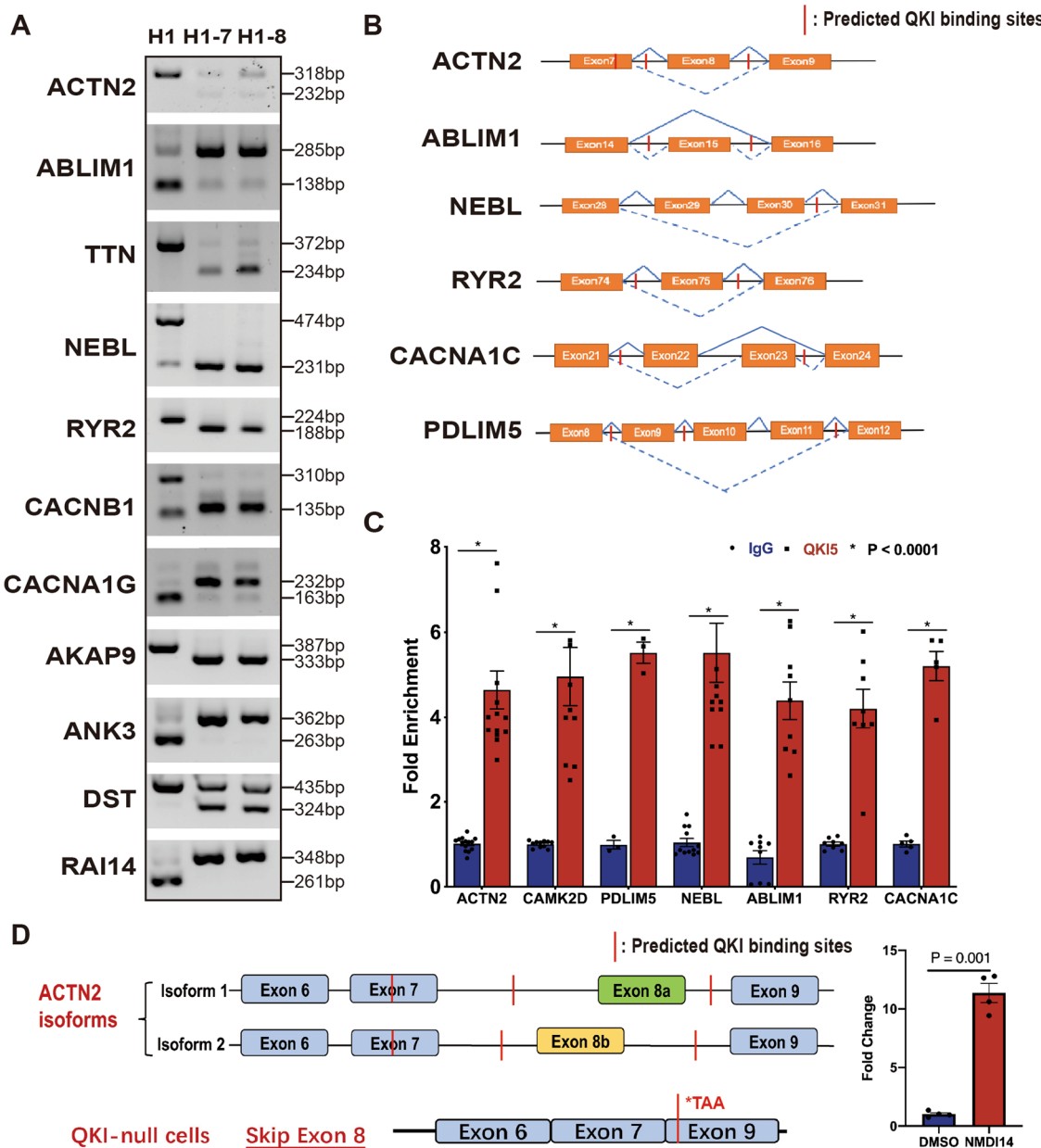

**Fig. 6 QKI-mediated alternative splicing events in genes critical to myofibrillogenesis. A** Representative RT-PCR confirmation of abnormal splicing events. All experiments are independently repeated at least three times with multiple independent sets of cell samples to confirm the reproducibility of the findings. **B** Schematic diagram of Skipped Exon events based on rMATs analysis. Solid lines indicated as normal splicing events, dash lines indicated as abnormal events in hCM-QKI^del. Relevant sashimi plots are in Supplementary Fig. 8. **C** RIP-qPCR analysis to verify the direct targets of QKI. Data were normalized to the IgG control group. Data represent the mean ± SEM from at least five independent experiments, statistical significance was determined by a two-tailed Student's t-test. **D** Schematic diagram to demonstrate a premature STOP codon (i.e., TAA) in ACTN2^SE-8 mRNA in H1-8 mutant cells. To confirm that the NMD pathway is responsible for the downregulation of ACTN2^SE-8 in H1-8 mutant cells, H1-8 cardiomyocytes were treated overnight with NMDI14 (25 μM), followed by qRT-PCR analysis on the level of ACTN2^SE-8 in H1-8 cardiomyocytes. Data are shown as the mean ± SEM, statistical significance was determined by a two-tailed Student's t-test.

consistent with its implicated role in RNA splicing. QKI-6 and QKI-7, which lack an NLS, were localized in cardiomyocyte cytoplasm and implicated in maintaining RNA stability and posttranscriptional modification. To further confirm whether hCM-QKI^del phenotype was due to QKI-5-mediated RNA-splicing activity or QKI-6/7-mediated regulation in RNA stability, or posttranscriptional modification, we used a PiggyBac (PB) transposon/doxycycline-inducible system to reactivate the expression of QKI-5, QKI-6, or QKI-7 in hESCs-QKI^del, respectively (Supplementary Fig. 11A). A total of three hESCs-QKI^del:

QKI-5^ind clones, four hESCs-QKI^del:QKI-6^ind clones, and three hESCs-QKI^del:QKI-7^ind clones were successfully established with confirmed doxycycline-induction-mediated QKI expression. Representative western blot screening is presented in Supplementary Fig. 11B.

We first tested hESCs-QKI^del:QKI-5^ind clones. Doxycycline induction was carried out at differentiation Day-0, -3, -6, and -8, and within 24 h, QKI-5 was successfully induced, as confirmed by western blottings. These hESCs-QKI^del:QKI-5^ind were differentiated into cardiomyocytes (hCMs-QKI^del:QKI-5^ind) and analyzed at D15

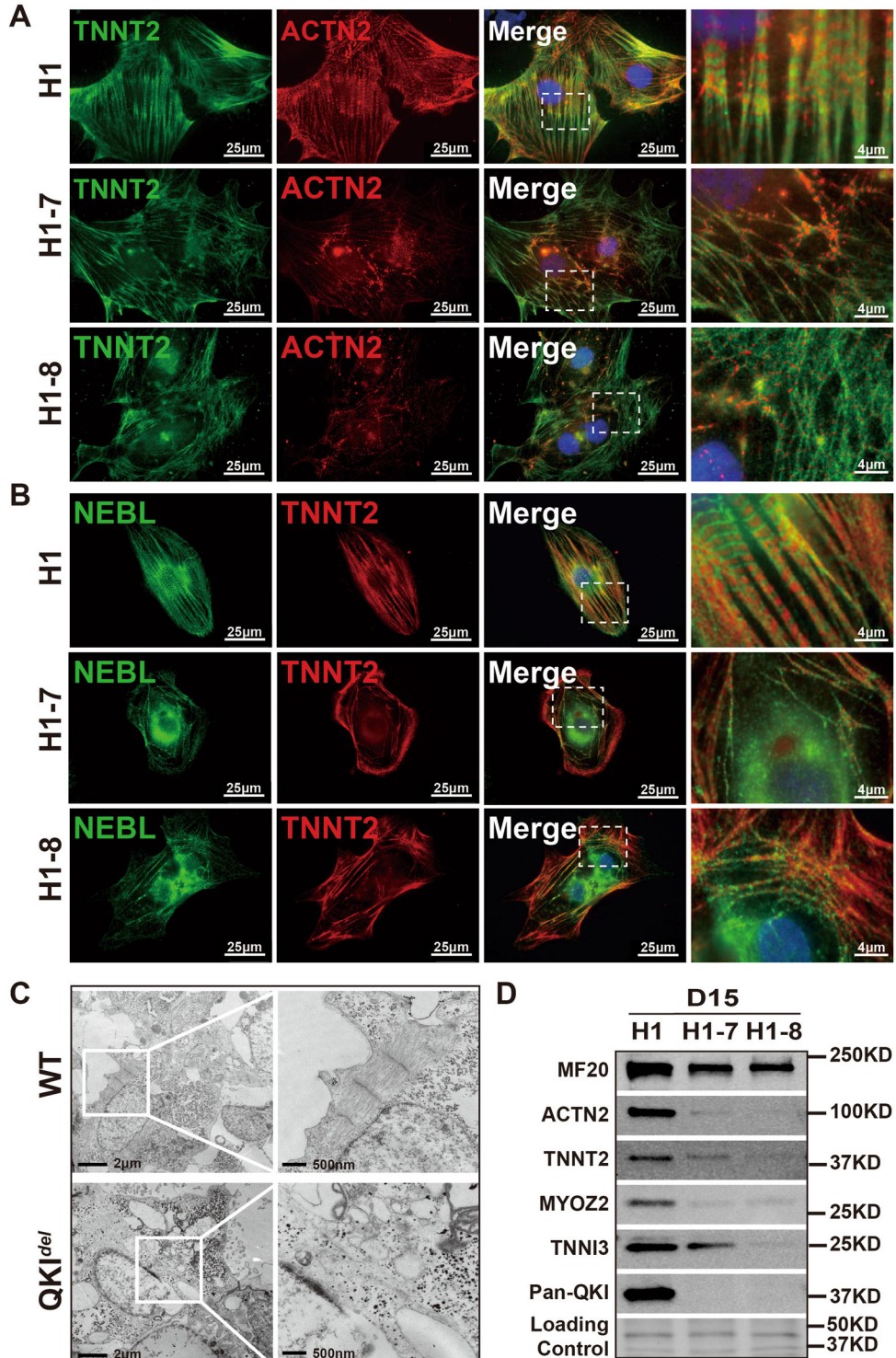

**Fig. 7 Severely altered myofibrillogenesis in hESCs-*QKI^del*-derived cardiomyocytes. A** Representative images of immunofluorescence staining of TNNT2 (green) and ACTN2 (red) in H1, H1-7, and H1-8 cardiomyocytes at differentiation Day-15. Scale bar: 25 μm. **B** Representative images of immunofluorescence staining of TNNT2 (red) and NEBL (green) in H1, H1-7, and H1-8 cardiomyocytes at differentiation Day-15. Scale bar: 25 μm. **C** Representative transmission electron microscopy images of normal and QKI-deficient differentiated cardiomyocytes at Day-15. Scale bar: 2 μm and 500 nm, respectively. **D** Western blot analysis of key myofibrillar proteins in control H1 and mutant H1-7 and H1-8 cardiomyocytes (Day-15). All experiments are independently repeated three times with three different sets of cell samples to ensure the reproducibility of the findings.

and D30. When QKI-5 was induced at days 0, 3, and 6 (but not at day-8), hCMs-*QKI^del*:*QKI5^ind* were able to be fully rescued to form well-organized beating cardiomyocyte sheets, similar to normal control cardiomyocyte sheets differentiated from hESCs (Supplementary Movies 7 and 8). Dual-immunofluorescence

staining demonstrated a full recovery of striated myofibril structure (Fig. 8) and the recovery of normal splicing events for all key genes examined in Day-15 differentiated cells (Supplementary Fig. 11C, D). This finding also suggested that QKI-5 was critical for the key transition stage when cardiac progenitors became early

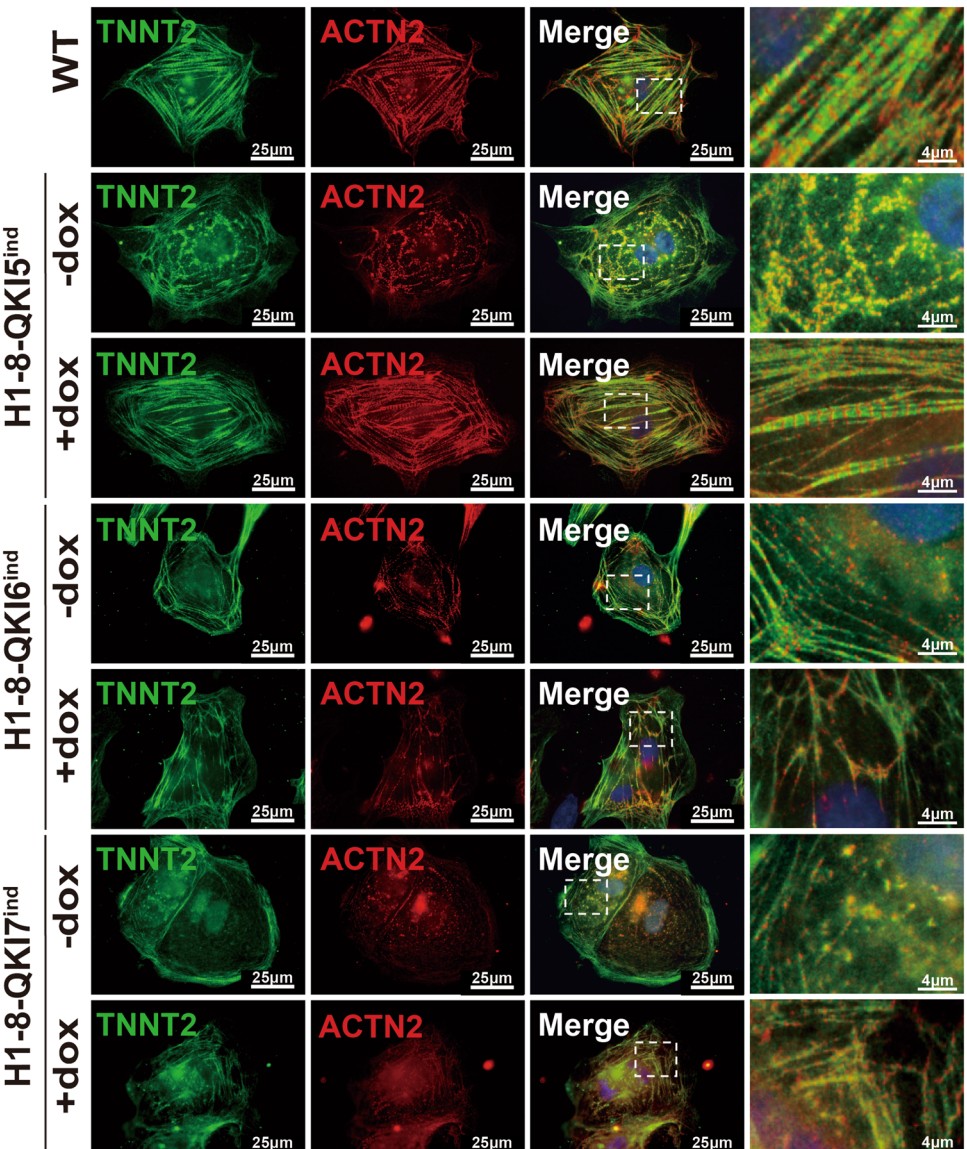

**Fig. 8 Reactivation of QKI5 in hESCs-*QKI^del* rescues myofibrillogenesis.** Inducible QKI5, QKI6, and QKI7 expression cells are established in H1-8 lines (H1-8-*QKI5^ind*, H1-8-*QKI6^ind*, H1-8-*QKI7^ind*). Representative images of immunofluorescence staining of TNNT2 and ACTN2 in H1-8-*QKI5^ind*, H1-8-*QKI6^ind*, H1-8-*QKI7^ind* cardiomyocytes with or without doxycycline induction. Myofibrillar structures are recovered in doxycycline-induced H1-8-*QKI5^ind* cardiomyocytes (Day-15) and ACTN2 expression is also recovered. However, doxycycline-induced H1-8-*QKI6^ind* and H1-8-*QKI7^ind* cardiomyocytes fail to recover ACTN2 expression and generate normal myofibrillar structures (Day-15). Scale bar: 25 μm. All experiments are independently repeated three times with three different sets of cell samples to confirm the reproducibility of the findings.

cardiomyocytes, which was consistent with the temporal expression pattern of QKI in the hESC-CM differentiation system and in mouse cardiovascular development. However, similar reactivation of QKI-6 and QKI-7 expression in hESCs-*QKI^del* failed to rescue the abnormal contractile phenotypes at both the functional and cellular levels (Fig. 8 and Supplementary Movies 7–10). The lack of striated myofibril structures and the lack of a normal form of *ACTN2* in hCMs-*QKI^del*:*QKI-6^ind* and hCMs-*QKI^del*:*QKI-7^ind* confirmed that the nuclear QKI-5 was the functional QKI isoform regulating the alternative splicing in developing cardiomyocytes. Collectively, these findings also suggested that QKI-6- and QKI-7-mediated activity in regulating RNA stability, post-transcriptional modification, or transportation were not essential to the cardiac myofibrillogenesis.

**Defect in myofibrillogenesis in *Qki*-deficient mice.** To confirm the above observations in vivo, we analyzed *Qki^βGeo/βGeo*

homozygous mutant embryos, which died in utero at E10.5 (Fig. 9A).[53] Histological analysis of E9.0-9.5 embryonic hearts demonstrated a thin ventricular wall with a significantly reduced level of trabeculation in *Qki^βGeo/βGeo* homozygous mutants relative to the level seen in the hearts of wild-type littermate or *Qki^βGeo/+* heterozygous embryos (Fig. 9B). *Qki^βGeo/βGeo* mutant hearts had similar downregulation of sarcomere genes (Fig. 9C and Supplementary Fig. 12A), which was consistent with the findings demonstrated by hCM-*QKI^del* scRNA-seq and bulk RNA-seq data. Dual-immunofluorescence staining using antibodies against cardiac α-actinin (*Actn2*) and troponin I (*Tnni3*) demonstrated dramatically reduced expression levels of *Actn2* and a significant defect in myofibrillogenesis in *Qki^βGeo/βGeo* hearts (Fig. 9D), which closely resembled the defect seen in hCMs-*QKI^del*. As we tested so far, all altered alternative splicing events in *Qki^βGeo/βGeo* mutants were identical to that found in

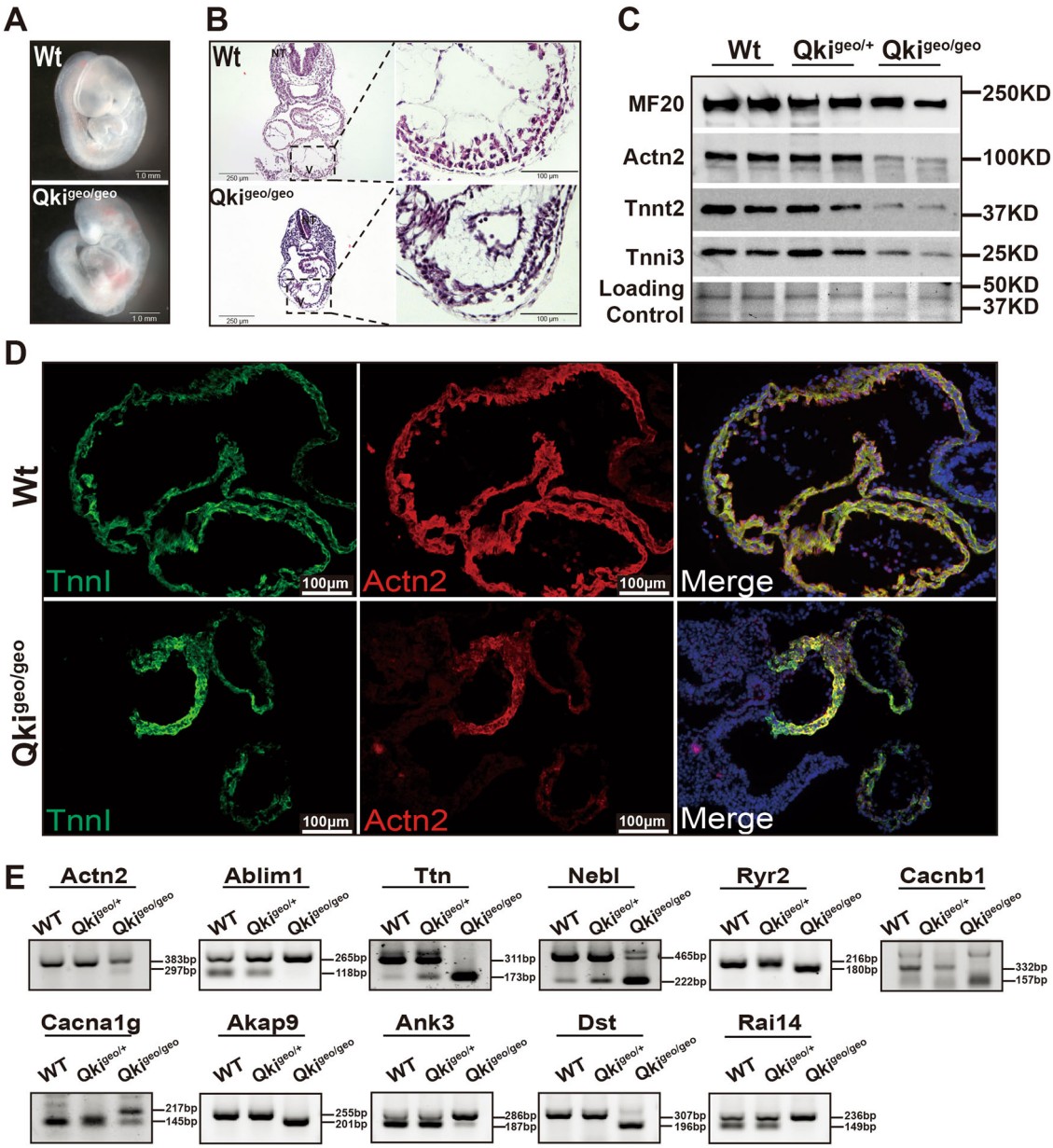

**Fig. 9 Defects in myofibrillogenesis in *Qki*^βGeo/βGeo^ mouse embryonic hearts. A** Compared to wild-type littermates, *Qki*^βGeo/βGeo^ embryos at E9.5 are growth compromised with significant pericardial effusion, suggesting embryonic heart failure. Scale bar: 1.0 mm. **B** Histological analysis of *Qki*^βGeo/βGeo^ and littermate control embryos and hearts demonstrates thin ventricular wall with a significantly reduced level of trabeculation in *Qki*^βGeo/βGeo^ ventricles. Scale bar: 100 μm. **C** Western blotting examination shows the expression of key sarcomeric proteins (myosin heavy chain, Actn2, Tnnt2, and Tnni3), which reveals significant downregulation of Actn2, Tnnt2, and Tnni3 in *Qki*^βGeo/βGeo^ hearts, which is similar to what was seen in cardiomyocytes derived from hESC-*QKI*^del^. **D** Representative images of immunofluorescence staining of TnnI3 and Actn2 in *Qki*^βGeo/βGeo^ and wild-type littermate control embryos at E9.0 show the severe defect in myofibrillogenesis in *Qki*^βGeo/βGeo^ hearts. Scale bar: 100 μm. **E** Representative RT-PCR results confirming alternative splicing events in *Qki*^βGeo/βGeo^ hearts (E9.25). All experiments are independently repeated at least three times with multiple different sets of samples to ensure the reproducibility of the findings.

hCMs-*QKI*^del^ (Fig. 9E), including the abnormal *Actn2* mRNA splicing variant lacking exon 8, containing a premature stop codon in exon 9, which was confirmed by sequencing analysis (Supplementary Fig. 12B). Collectively, these findings validated the critical role of *Qki* in cardiac myofibrillogenesis and contractile physiology in vivo.

## Discussion

Pre-mRNA alternative splicing is one of the most important biological features in eukaryotes[65]; this process provides an additional mechanism for posttranscriptional regulation of gene functions at the pre-mRNA level[66]. As we show here, QKI is an important RBP that is highly expressed in developing hearts and adult hearts. By using both the hESC-cardiomyocyte differentiation system and mouse genetic model, we are able to demonstrate four major findings in our current study: (1) despite its expression in pluripotent embryonic stem cells, QKI does not appear to have important functions in the maintenance of pluripotency, the ability to self-renew and early differentiation towards early cardiogenic progenitors; (2) QKI is essential for differentiating cardiomyocytes during the transition from cardiogenic progenitors to early cardiomyocytes when important myofibrillogenesis and

contractile physiology are initiated in the transition to a fully functional state; (3) QKI safeguards proper alternative splicing events critical to genes contributing to myofibrillogenesis in developing cardiomyocytes; and (4) QKI-5 is the major QKI isoform in mediating pre-mRNA alternative splicing in cardiomyocytes.

*QKI* has drawn great attention for its potentially important disease-causing contribution to various human diseases, including schizophrenia[67], 6q terminal deletion syndrome[68], myelin disorders[69], and cancers[41,70,71]. Interestingly, a recent two-stage genome-wide association study suggested that QKI was associated with myocardial infarction and coronary heart disease[72]. In addition, it was shown that QKI-5 and QKI-6 expression was associated with hypoxia/reoxygenation and ischemia/reperfusion (I/R) injuries in cardiomyocytes and heart[50,73]. The over-expression of QKI-5 or QKI-6 in neonatal cardiomyocyte could prevent I/R-induced cardiomyocyte death[50]. Although the underlying mechanism of this association was largely elusive, the potential role of QKI in regulating vascular smooth muscle dedifferentiation in response to vessel injuries was demonstrated in a mouse model, in which *QKI* functioned as a splicing regulator of a key smooth muscle transcription factor, myocardin[55]. As myocardin was also a key transcription factor in early cardiomyocyte differentiation[74,75], we have particularly focused on potentially altered myocardin splicing in our differentiating hESCs-*QKI^del*. However, we did not find significant alterations in myocardin alternative splicing during cardiomyocyte differentiation, suggesting a context-specific function of QKI. This notion also identifies a critical question as to what the role of QKI plays in the differentiation of various cellular lineages. Although our current study is not able to address this question, the hESC-*QKI^del* mutant lines will serve as an important resource for addressing these questions in the future.

One of the major findings in our study is that QKI plays a critical role in regulating alternative splicing for genes involved in Z-disc formation, which is a critical step in sarcomerogenesis. The Z-disc, also called Z-line, provides an anchor for the sarcomere structure. ACTN2, NEBL, ABLIM1, and PDLIM5 are critical components of the Z-disc, and TITIN connects the Z-disc to the M-line in the sarcomere. Importantly, hCMs-*QKI^del* demonstrate significantly altered alternative splicing events in these Z-disc genes. As confirmed by the RIP-qPCR assay, these Z-disc genes are direct targets of QKI. Interestingly, the skipping of exon 8 in *ACTN2* also leads to the generation of a premature stop codon, which subsequently leads to the dramatic downregulation of *ACTN2* via the NMD pathway. Collectively, our data demonstrate the critical safeguard function of QKI in ensuring normal processing of pre-mRNA splicing for generating a functional myofibril structure suitable for contractile physiology in cardiomyocytes. In addition to Z-disc genes, it is worth noting that many other altered alternative splicing events are found in hCMs-*QKI^del*. For example, there was noticeable alternative splicing of genes involved in regulating calcium dynamics and E–C coupling, which is among the most significantly altered events in hCMs-*QKI^del*, including the voltage-gated calcium channel (e.g., CACNA1C, CACNA1G), calcium release channel (e.g., RYR2), and CAMK2D. This alteration likely contributes to the significant delay of E–C coupling in hCMs-*QKI^del* (Fig. 2C, D). This notion suggests that QKI is also involved in multiple critical components essential for cardiac physiology and pathophysiology. Additional systematic analysis in the future should be given to these unique events, which will help broaden our knowledge of the role that QKI plays in cardiovascular development and function and will help identify its potential involvement in the pathogenesis of cardiomyopathies and cardiovascular diseases.

Interestingly, although our analysis was solely based on cardiomyocytes, many altered alternative splicing events were found in genes highly relevant to neuronal physiology, as demonstrated by the transcriptome analysis (Figs. 4 and 5). There appear many shared QKI-mediated alternative splicing events in the nerve system and cardiomyocytes. Interestingly, a large-scale genome-wide analysis of patients with congenital heart defects demonstrated that a similar candidate gene profile contributes to both congenital heart defects and neurodevelopmental and functional defects[76,77]. It is of great interest to test whether these altered splicing events share common pathways in contributing to both defects in cardiovascular development and neurodevelopment in the future.

In summary, our work has revealed the role of *QKI* in cardiogenesis and heart function. Our data demonstrate that *QKI* is a critical alternative splicing regulator in both human and mouse cardiomyocytes, which is not only indispensable for cardiogenesis but also essential to heart function. Our work finds that QKI is potentially involved in the pathogenesis of certain forms of cardiovascular diseases.

## Methods

**Generation of QKI-null hESC lines**. hESC lines H1, H7, and H9 were purchased from WiCell. To generate the QKI-null hESC line, we utilized the CRISPR/CAS9 genome-editing tool, using the D10A-mutant nickase version of Cas9 (Cas9n) with a pair of offset sgRNAs complementary to opposite strands of the target site. We designed sgRNAa and sgRNAb targeting exon 3 of the human QKI gene (http://crispr.mit.edu:8079/). pX462-sgRNAa and pX462-sgRNAb were transfected into hESCs by electroporation (Neon Transfection System, ThermoFisher Scientific). The QKI-null hECS clones were screened by western blot analysis after 1 week of puromycin selection and the positive clones were further validated by genomic sequencing.

***Qki^βGeo* mice**. *Qki^βGeo* mouse line was previously generated[53]. Our experimental mice were housed in an AAALAC (Association for Assessment and Accreditation of Laboratory Animal Care)-approved animal facility in Indiana University School of Medicine, an biosafety level 2 facilty with temperature and humidity controlled at 75°F and 50%, respectively, and with light/dark cycle set at 14 h light/10 h dark. All procedures for animal handling, housing, and mouse embryo and tissue harvesting were conformed to the US National Institutes of Health Guide for the Care and Use of Laboratory Animals (NIH Publication No. 85-23, revised 1996) and were approved by IACUC (Institutional Animal Care and Use Committee) of Indiana University Purdue University Indianapolis under protocol #20064.

**Generation of doxycycline-inducible QKI expression hESC-QKIdel lines**. We used a PB transposon system to generate doxycycline-inducible expression of QKI5, QKI6 and QKI7 in hESC-*QKI^del* lines (Supplementary Fig. 9A). The QKI5, QKI6, and QKI7 cDNA fragments were inserted into a PB-TRE3G-EGFP-PGK-Neo vector between the BamHI and BsrGI sites. These vectors were cotransfected with a Hyper-PB vector and a PB-CAG-rtTA-IRES-puro vector into hESC-QKI^del lines by electroporation. Electroporated cells underwent one week of puromycin selection and neomycin selection for another week. The positive clones that expressed QKI5, QKI6 or QKI7 under the induction of doxycycline were screened by western blot analysis.

**hESC culture and cardiomyocyte differentiation**. hESCs were cultured in mTesR medium (STEMCELL #85852). hESCs were differentiated into cardiomyocytes following previously published protocols[78]. In brief, the hESCs were dissociated using Accutase (Millipore Sigma, A6964) and were cultured on Matrigel-coated plates in mTesR medium. Cardiomyocyte differentiation was initiated when cells were ~95% confluent by culturing cells in RPMI1640/B-27 minus insulin medium RPMI1640 (GIBCO) with B-27 supplement minus insulin (ThermoFisher Scientific, A1895601) supplemented with 6 μM CHIR-99021 (Selleckchem, S1263) for differentiation on days 0 and 1. For days 3 and 4, IWP-2 (3 μm, Tocris, 3533) was added to the fresh medium. From day 5 onward, cells were cultured with fresh RPMI1640/B-27 medium every 2 days. A beating cluster of cells appeared ~7–8 days post differentiation and robust spontaneous-beating sheets occurred by day-10. Cultures were maintained in Dulbecco's modified Eagle's medium with 10% fetal bovine serum in a 37 °C and 5% $CO_2$ air environment.

**Immunofluorescent analyses of cultured cells and embryonic tissues**. Individual hESC clones and single cells generated from beating sheets were plated on 15 mm Matrigel-coated circular coverslips. Cells were fixed and permeabilized with cold methanol, which was followed by incubation with 5% BSA for blocking before

the addition of antibodies for staining. To assess pluripotency, hESC colonies were stained with antibodies against OCT3/4 (Stemgent, 09-0023), SSEA-4 (Thermo-Fisher Scientific, 41-4000), SOX2 (Stemgent, 090-0024), and TRA-1-60 (Stemgent, 09-0010). To assess differentiated cardiomyocytes, cells were stained with antibodies against TNNT2 (Abcam, ab45932), ACTN2 (Sigma, A7811), NEBL (ThermoFisher Scientific, PA5-53106), and TTN (DSHB, 9D10). Anti-QKI-5 antibody (Bethyl; A300-183A) was used in the immunofluorescence staining to confirm our successful ablation of QKI in hESCs. All first antibodies were diluted at 1 : 100 or at a concentration specified by manufactures. Corresponding secondary antibodies labeled with either Alexa Fluor 594 or 488 were applied at a dilution of 1 : 1000 (Donkey anti-Mouse IgG-594: ThermoFisher Scientific, A21203; Goat anti-Rabbit-594: ThermoFisher Scientific, A11012; Donkey anti-Goat-488: Thermo-Fisher Scientific, A11055; Goat anti-Mouse IgG: ThermoFisher Scientific, A11001; Goat anti-Rabbit IgG: ThermoFisher Scientific, A11008) to visualize the immunoreactivities. Then, cells were subsequently counterstained with 4′,6-diamidino-2-phenylindole (DAPI) and sealed with a prolonged antifade reagent (ThermoFisher Scientific, P36931).

To determine the histological structure of developing hearts, mouse embryos were fixed, embedded in OCT (Sakura Finetek, Japan), frozen, and cut to produce 10 μm-thick sections, which was followed by the standard procedure of antibody staining, including the use of a brief fixation with 2% paraformaldehyde (PFA) for 5 min, washing with PBST (PBS + 0.1% Triton X-100), treating with M.O.M. mouse IgG blocking reagent (Vector laboratories), incubation with primary antibody (diluted in M.O.M. reagent) and incubation with secondary antibodies with Alexa Fluor 594 or 488. The slides were counterstained with DAPI and sealed with prolonged antifade reagent.

**Quantitative RT-PCR and western blot analyses**. Total RNA was isolated using the Trizol reagent (ThermoFisher Scientific) according to the manufacturer's instructions. RNA (1 μg) was reverse-transcribed into cDNA by using the iScript Reverse Transcription Supermix for RT-qPCR (BIO-RAD). Quantitative RT-RCR was performed with the PowerUp SYBR Green PCR Master Mix (ThermoFisher Scientific, A25742) with RPL7 as an internal control. Day-15 differentiated CM cells were incubated with NMDI14 (25 μm), an inhibitor of nonsense-mediated mRNA decay, for 18 h. RNA was extracted and the relative *ACTN2* expression level was determined by RT-PCR. *ACTN2* mRNA levels were normalized to *TNNT2* levels and are displayed relative to the DMSO control sample (DMSO, defined as 100%). The primer sequences are listed in Supplementary Table 1.

Total protein was extracted from cultured cells or isolated embryonic hearts using RIPA lysis buffer supplemented with a protease inhibitor cocktail (Sigma). Equal amounts (30 μg) of protein were boiled in 1× SDS loading buffer for 10 min and separated on a 4-20% Mini-PROTEAN TGX Stain-Free Gels (Bio-Rad, 4561093), which was followed by transfer to nitrocellulose membranes (Bio-Rad, 1704158) using the Trans-Blot Turbo Transfer System. The membranes were blocked with 5% nonfat dry milk (Bio-Rad, 1706404) in Tris-buffered saline containing Tween-20 and incubated with primary antibodies against TNNT2 (DSHB, Cat#AB528495), ACTN2 (Sigma, Cat#A7811), MF20 (DSHB, Cat#AB2147781), MYOZ2 (ThermoFisher, Cat#PA5-76946), TNNI3 (Santa Cruz, Cat#SC15368), Tubulin (Sigma, Cat#T6199), Pan-QKI(Abcam, Cat#ab126742), QKI-5 (Bethyl, A300-183A), QKI6 (Millipore, Cat#AB9906), and QKI7 (Millipore, Cat#AB9908), respectively, at 1 : 1000 dilution, and corresponding secondary antibodies labeled with horseradish peroxidase (Goat anti-Mouse IgG antibody, ThermoFisher Scientific, G21040; Mouse anti-Rabbit IgG antibody: Santa Cruz, sc-2357) at 1 : 4000 dilution. Immunoreactive bands were visualized by ECL (Thermo Scientific, 32106), which was developed and quantified using a gel documentation imaging system (GelDoc and Image Lab, BIO-RAD).

**Transmission electron microscopy**. Cultured differentiated cardiomyocytes were fixed with 2.5% (vol/vol) glutaraldehyde for 15 min at room temperature and post-fixed with 1% osmium tetroxide for 2–3 h. The samples were dehydrated by incubation in graded ethanol (50%, 70%, 90%, and 100%) and in propylene oxide for 10 min. Samples were embedded, sectioned at 70-nm thickness, and stained with lead citrate. Images were captured using a PHILIPS CM-120 TEM (PHILIPS, Holland).

**Single-cell 3′ RNA-seq and bioinformatics analysis of scRNA-seq**. Single-cell 3′ RNA-seq experiments were conducted using the Chromium single-cell system (10× Genomics, Inc.) and Illumina sequencers at the Center for Medical Genetics of Indiana University School of Medicine. Single-cell suspensions were carefully prepared by digesting differentiated cell sheets with collagenase I (1 mg/ml, Sigma) for 60 min, and then incubating the cells in 0.25% trypsin without EDTA for 10 min at 37 °C. Cell suspensions were filtered with a 40-μm cell strainer (BD Falcon). Day-6 and Day-15 single-cell suspensions were first inspected on the Countess II FL (ThermoFisher Scientific) and under microscope for cell number, cell viability, and cell size. Depending on the quality of the initial cell suspension, the single-cell preparation included centrifugation, re-suspension, and filtration to remove cell debris, dead cells and cell aggregates. Single-cell capture and library preparation were carried out according to the Chromium Single cell 3′ Reagent kits V2 User Guide (10× Genomics PN-120267, PN-1000009, PN-120262). Appropriate number of cells were loaded on a multiple-channel micro-fluidics chip of the Chromium

Single Cell Instrument (10× Genomics) with a targeted cell recovery of 10,000. Single-cell gel beads in emulsion containing barcoded oligonucleotides and reverse transcriptase reagents were generated with the v2 single-cell reagent kit (10× Genomics). Following cell capture and cell lysis, cDNA was synthesized and amplified. Illumina sequencing libraries were then prepared with the amplified cDNA. The resulting libraries were assessed with an Agilent TapeStation or Bioanalyzer 2100. The final libraries were sequenced using a custom program on Illumina NovaSeq 6000. Twenty-six basepairs of cell barcode and unique molecular index (UMI) sequences, and 91 bp RNA reads were generated.

CellRanger 2.1.0 (http://support.10xgenomics.com/) was utilized to process the raw sequence data generated. Briefly, CellRanger used bcl2fastq (https://support.illumina.com/) to demultiplex raw base sequence calls generated from the sequencer into sample specific FASTQ files. The FASTQ files were then aligned to the human reference genome GRCh38 with RNA-seq aligner STAR. The aligned reads were traced back to individual cells and the gene expression level of individual genes were quantified based on the number of UMIs detected in each cell. The filtered gene-cell barcode matrices generated with CellRanger were used for further analysis with the R package Seurat (version 2.3.1) with Rstudio version 1.1.453 and R version 3.5.1[79]. QC of the data was implemented as the first step in our analysis. We first filtered out genes that detected in less than five cells and cells with less than 200 genes. To further exclude low-quality cells in downstream analysis we used the function is Outlier from R package scater together with visual inspection of the distributions of number of genes, UMIs, and mitochondrial gene content[80]. Cells with extremely high or low number of detected genes/UMIs were excluded. In addition, cells with high percentage of mitochondrial reads were also filtered out. After removing likely multiplets and low-quality cells, the gene expression levels for each cell were normalized with the NormalizeData function in Seurat. To reduce variations sourced from different number of UMIs and mitochondrial gene expression, we used the ScaleData function to linearly regress out these variations. Highly variable genes were subsequently identified.

To integrate the single-cell data of the control and QKI-deficient samples, functions FindIntegrationAnchors and IntegrateData from Seurat v3.1.0 were implemented.[58] The integrated data were then scaled and PCA was performed. Seurat functions FindNeighbors and FindClusters were applied for shared nearest-neighbor graph-based clustering. The FindConservedMarkers function was subsequently used to identify canonical cell-type marker genes. Cell cluster identities were manually defined with the cluster-specific marker genes or known marker genes. The cell clusters were visualized using the UMAP plots. To compare average gene expression within the same cluster between cells of different conditions, we applied the function AverageExpression. R packages ggplot2 (http://ggplot2.org) and ggrepel (https://github.com/slowkow/ggrepel) were used to plot the average gene expression. Violin plots were used to visualize specific gene expressions across clusters and different sample conditions. To infer the developmental trajectories from the data, Monocle 3 was utilized[59]. The Seurat object from the integrative analysis was converted to SingleCellExpression object using the function as.SingleCellExperiment in Seurat. The SingleCellExpression object was then used to generate Monocle 3 CDS object. The gene-cell count matrix, dimensional reductions and clusters determined from the Seurat analysis were directly imported into the Monocle CDS object. Trajectory graph was learned from the CDS object and the cells were ordered in pseudotime along the learned trajectory. Cells of early progenitors were used as root cells in the order_cells process. All single-cell RNA sequence data that support the findings of this study have been deposited in Geo Database (GSE144009).

**Bulk RNA-seq and alternative splicing analysis**. Total RNA samples from cultured cells were prepared using a RNeasy Plus Micro Kit (Qiagen) according to the manufacturer's instructions. The RNA-seq library was prepared according to the manufacturer's instructions (KAPA mRNA Hyperprep Kit (KK8581)). The cDNA library was sequenced on an Illumina sequencing platform (Illumina HiSeq™ 4000, San Francisco, USA) using paired-end technology. The sequence reads were mapped to "UCSC reference genome hg38" for expression analysis and "Ensembl GRCh38v95" for Spliced Transcripts Alignment reference[81]. To evaluate the quality of the RNA-seq data, the number of reads that fell into different annotated regions (exonic, intronic, splicing junction, intergenic, promoter, UTR, etc.) of the reference genes were determined with bamUtils[82]. Low-quality mapped reads (including reads mapped to multiple positions) were excluded and feature Counts was used to quantify the gene level expression[83]. Differential gene expression analysis was performed with edgeR[84]. In this workflow, the statistical methodology applied used negative binomial generalized linear models with likelihood ratio tests. The bulk RNA-seq data have been deposited in Geo Database (GSE144008).

Splicing analysis was conducted using rMATS (version 3.2.5), in which the percent-spliced-in (PSI) for each splicing event were calculated based on the number of sequencing reads supporting the inclusion and exclusion isoforms, respectively[85]. A FDR and the changes on the exon inclusion levels (deltaPSI) were calculated for each event based on four biological replicates in each condition. Sequences for the primers used to confirm alternative splicing are provided in Supplementary Table 1.

**Whole-mount X-Gal staining and histological analysis of mouse embryos**. Mouse embryos were fixed for 15 minutes at room temperature and washed in

wash buffer three times for 15 min. Embryos were placed in X-Gal staining buffer from 4 h to overnight at 37 °C, depending on the level of LacZ activity. X-Gal-stained embryos were fixed with 4% PFA, paraffin-embedded, cut into 10 μm-thick sections, and subjected to nuclear fast red staining.

**RNA immunoprecipitation**. RIP experiments were conducted using an EZ-Magna RIP RNA-Binding Protein Immunoprecipitation Kit (Millipore). Human QKI5 antibody was used to pull down RNAs. Normal mouse IgG (provided by kit) was used as a negative control. Total RNA was purified using the RNeasy Plus Micro Kit (Qiagen) according to the manufacturer's instructions.

**Statistical analysis**. Experimental data were reported as the mean ± SEM, unless otherwise indicated. Two-tailed Student's $t$-tests were used when comparing the differences of two groups, and one-way analysis of variance test was used to compare among three groups. $P$-values < 0.05 were considered significantly different. To ensure the reproducibility of our data, all experiments were repeated at least three times with multiple sets of independent samples. GO enrichment and KEGG pathway enrichment analysis of differentially expressed genes were performed respectively using R programming based on the hypergeometric distribution.

**Reporting summary**. Further information on research design is available in the Nature Research Reporting Summary linked to this article.

## Data availability

If needed, contact N.S. for original data described in Fig. 7C and Supplementary Fig. 3D, E, and contact W. Shou for all other original data described in the paper. Contact K.Y. for requesting Qki$^{βGeo}$ mouse strain, N.S. for requesting hESCs-$QKI^{del}$ (H7) line and hESCs-$QKI5^{ind}$ (H7) line, J.N. for requesting the plasmids of inducible PiggyBac transposon overexpression system, and W.S. for requesting all other reagents described in this article. The bulk RNA Sequence data have been deposited in GEO Database (GSE144008) and single-cell RNA sequence data that support the findings of this study have been deposited in GEO Database (GSE144009). In addition, the gene ontology (GO) and Kyoto Encyclopedia of Genes and Genomes (KEGG) database used in the study are available at [http://geneontology.org/] and [https://www.genome.jp/kegg/]. Source data are provided with this paper.

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

## Acknowledgements

We thank all members in Shou lab, Sun Lab, and Huang lab for critical discussion and proofreading. This work was supported in part by National Institute of Health P01HL134599 and R01HL145060, and Riley Children Foundation (to W.S.). N.S. was supported by the National Natural Science Foundation of China #31571527; X.C. was supported in part by visiting scholar fund from China Scholarship Council (CSC); C.X. was supported in part by visiting scholar funds from CSC and the Young Investigator Award by the National Natural Science Foundation of China (#81500241), and fund from the Innovative Research Team of High-Level Local Universities in Shanghai; W. Sheng was supported by the fellowship from Haiju program of National Children's Medical Center (EK1125180102).

## Author contributions

X.C., Ying Liu, and C.X. carried out the majority of the experiments, performed data analyses, and generated data Figures and Supplementary Information. X.C. provided first draft of the manuscript. Ying Liu and C.X. participated in revising the manuscript; L.B., Z.L., X.L., and J.H. contributed significant roles in technique assistance. E.S., H.G., and Yunlong Liu contributed significant roles in scRNA-seq and bulk RNA-seq and sub-sequent bioinformatics analysis. W. Sheng, H.Q., H.J., and M.S. contributed to technique support for experiments. D.C., C.C., L.Y., X.L., J.N., and K.Y. contributed to the work for providing key reagents and technique guidance. G.H., W.S., and N.S. conceptually conceived and initiated the work, and finalized the manuscript. W. Shou served as the lead contact for the work.

## Competing interests

The authors declare no competing interests.
