## [Peer Review File · Nature Communications]

Reviewers' comments:

Reviewer #1 (Remarks to the Author):

The authors have identified the RNA binding protein Quaking (Qki) as a key regulator of cardiomyocyte differentiation and function. Their sequencing analysis includes single-cell interrogation of cardiomyocytes differentiated from wildtype and Qki-deficient hESCs, as well as bulk RNAseq of differentiated cardiomyocytes. Their findings reveal that Quaking does not have a large impact on early cardiomyocyte specification (nor does it affect the stemness of the undifferentiated hESCs), but rather confers a significant defect in the functionality of differentiated cardiomyocytes. It is shown that differentiated cardiomyocytes which lack Qki have a suite of downregulated genes involved in sarcomere function, thus explaining the defect in differentiated cardiomyocytes. Further analysis revealed that the nuclear isoform of Qki, Qki5, is responsible for regulating a network of alternatively spliced genes which play an important role in the functionality of differentiated cardiomyocytes. Finally, a knockout mouse model shows severe defect in myofibrillogenesis and share similar defects in splicing as the cell lines.

Comments

1. Figure 1: The RT-PCR and WB should be performed for each QKI isoform. QKI-5, QKI-6 and QKI-7. Pan antiQKI and 'qki' mRNA is not useful as to define which isoform is likely involved.

1.1. Figure 1B, authors should add molecular weight marker. Pan-Qki antibody is used and only one band is shown, however due to the difference in sizes of the Qki isoforms -6 and -7, should we be seeing multiple bands? Western blots of specific Qki isoforms would be helpful here (see comment 1), even though rescue experiments with individual isoforms are provided later on.

2. Figure 2: The finding that H1-7 and H1-8 clones had 20% reduction in TNNT2-positive cells suggests that the differentiation defect is partially due to a loss in the number of differentiated cells. However, this does not fully explain the drastic defect in cardiomyocyte differentiation. Therefore, the scRNA analysis is useful in determining whether the defect originates in the process of differentiation. The authors argue that the scRNA analysis shows there is little difference in early lineage specification between Qkidel and WT hESCs. However, it would be much easier to visualize this claim if the authors could provide pseudotemporal trajectories of Qkidel and WT hESCs using the

Monocle ordering algorithm. This would more clearly show that the differentiation defect observed in Qkidel cardiomyocytes is not due to early specification/determination.

3. Further, it would be expected that the differentiated cardiomyocyte population (described as being represented by clusters 0-3 at d15), should have a 20% reduction in cell number compared to the WT. Is this reflected in the scRNA analysis? This would corroborate the flow cytometry findings of reduced differentiated cell number.

The scRNAseq analysis reveals no defect in the differentiation pathway, so the 20% reduction in the differentiated population remains to be addressed, are they undergoing apoptosis?

4. Results section "QKI-mediated alternative splicing in cardiac myogenesis": Galarneau and Richard 2005 should be referenced here when discussing the Quaking consensus sites. Supplemental figure 6 requires a bit more explanation, what target sequences were assessed for Qki binding sites? Did the authors focus only on regions which were alternatively spliced, or the full transcript? It could also be mentioned in the text for clarity that, according to the data presented in supp figure 6, the Qki core site (ACUAAAY) was enriched, while the half-site (UAAY) was not significantly enriched compared to the background sequences.

5. Figure 5 Panel B: Quaking binding site locations should be indicated in these splicing diagrams.

6. Panel D: RIP-qPCR shows that Qki binds to ACTN2, but an assay that would provide a more precise binding location such as ParCLIP should be done to confirm that Quaking binds to the specific Quaking binding sites which flank exon 8.

7. Figure 6. Panel D: The downregulation of sarcomeric proteins whose RNA levels and splicing are unaffected by Qki deficiency (presumably, an example would be MYOZ2 which is not mentioned elsewhere in the context of RNA but shows drastic reduction in protein levels), may be a symptom of the overall defective differentiated state of these cardiomyocytes, however it is also possible that Qki is regulating of RNA stability of these targets. It may be worthwhile to check the 3' UTR sequences of these sarcomeric transcripts for the presence of Qki binding sites to assess whether the functionality of Qki as a regulator of RNA stability is relevant here.

8. Also, the WB of d15 cardiomyocytes shows that TNNT2 protein is almost completely gone in the Qki deficient cells. However, earlier flow cytometry analysis (supp fig 3) shows that there is only a 20% reduction in TNNT2-positive cells in Qki deficient d15 cardiomyocytes. Did the immunofluorescence analysis confirm the flow cytometry data by showing less TNNT2-positive cells? There is a lack of consistency here that should be addressed.

9. Supp Fig 10 Which antibodies are used for Qki isoforms? Please provide details in materials and methods. Qki-5, -6, and -7 are shown as all being 37kDa but should have different molecular weights, again specific labeling of molecular weight markers is important here. Western blots for Qki 6 and 7 should be included as control for Qki5 OE in Qkidel hESCs. Authors should specify what day of differentiation alternative splicing recovery (panel C) was done.

10. Figure 7 Very convincing findings that Qki5 is the isoform responsible for the phenotype. However, the finding that Qki5 induction only works at early stages of differentiation (days 0-6 but not day 8) to correct the differentiation defect contradicts earlier findings that Qki does not impact early specification of cardiomyocytes and is only involved at later, terminal stages and in the functionality of differentiated cardiomyocytes. This finding would imply that Qki does indeed play a role in early differentiation, although the effects may not be observable until terminal differentiation. This needs to be addressed.

11. Supp figure 11 It should be noted that the alternative splicing of ACTN2 is not as striking in the Qkigeo/geo embryonic heart as was seen in the cell line, I would refrain from using the term "identically altered". Other targets shown earlier (fig 5a) should also be shown here to confirm similar alternative splicing patterns.

12. The discussion of alternative splicing in the introduction is lacking appropriate references. Info required regarding the QkiBgeo mice should also appear in materials and methods

13. This sentence is inaccurate as written. QKI does not have SH2 and SH3 domains. "QKI contains an RNA-binding motif (KH domain) and protein interaction and signaling domains (SH2 and SH3 domains), which are flanked by QUA1 and QUA2 domains."

14. "As we shown here, QKI is a novel RNA-binding protein that is highly expressed in developing hearts and adult hearts." Remove novel as QKI is not a novel RBP.

15. The QUA domain is involved in forming homo- or heterodimers and is required for RNA binding.^{32, 33, 34, 35} Chen and Richard 1998 MCB 18:4863 should be cited instead of Darbelli and Richard 2016.

16. QKI5 has been implicated for its major function in pre-mRNA splicing regulators,^{37, 38, 39}

Delete reference 37 – they perform parCLIP to identify intronic sequences. they do not perform alternative splicing experiments

These should be cited here.

Wu et al. PNAS 99:4233

Zong et al., PloS Genet 2014 e1004289

Darbelli et al., J Neuroscience 2016

de Bruin et al., Nat Neurosci 2016

while QKI6 and QKI7 appear to be more relevant to RNA-posttranscriptional processing and transportation.³³ Primary references should be cited. here.

Reviewer #3 (Remarks to the Author):

This is an interesting study about the role of a RNA binding protein QKI in cardiac development. I will focus my review on the aspects related to scRNA-seq.

This is a very nice scRNA-seq analysis and the authors have made it clear about what they have done in the Methods section. Kudos to them.

I am curious by the many cardiac subpopulations the authors have found in their cultures: 8 cardiac clusters in day 6 cells and 4 clusters in day 15 cells. Do they represent different stages of cardiac differentiation? (Day 6 clusters 0, 4 and 7 appear to express less TNNT2 than the other cardiac clusters – are they more toward the progenitor stage?) Or are they related to different lineages like first vs. second heart fields, atrial vs. ventricular? I am interested to see the differences among these clusters. A cell fate trajectory analysis like Monocle might also be helpful.

QKI KO cells may adopt alternative cell fates, as suggested by a reduction in cTnT+ cells by day 15. Therefore, I expect to see some clusters that primarily consist of KO cells. However, it appears to be

the other way around: cluster 8 in day 6 and cluster 6 in day 15 are mostly from wildtype cells. They don't seem to be cardiac. What are they?

The authors used 40 um strainers for their cardiomyocyte prep. I assume the wildtype cardiomyocytes are more mature and perhaps bigger since they beat better. Will the 40 um strainer be more likely to filter out the more matured cardiomyocytes? If so, there might be even a bigger difference between wildtype and KO cardiomyocytes.

Review: RNA-binding protein QKI is a critical pre-mRNA alternative splicing regulator of cardiac myofibrillogenesis and contractile function

Evidence that the RNA-binding protein, QKI, plays a crucial role in myofibril formation and function is supported in convincing detail in the submitted manuscript. The **expression in vitro** of key myofibril proteins in QKI-deficient cells and in human control embryonic stem cells (hESCs), in the **cardiomyocyte differentiation** system **provided a clear view of the differences** in myofibrillogenesis **seen** in QKI-deficient cells when compared with QKI control cells. The immunofluorescent imaging shown in the Figures convincingly demonstrates that, in QKI-deficient cells, there is a notable loss of **aligned** myofibers, together with a fractured morphology of the fibers, in place of the parallel bundles of myofibrils formed during cardiomyocyte differentiation in control hESCs. The distribution of sarcomere proteins along the myofibers, additionally, is shown to be disrupted in mutant QKI cells, eliminating the characteristic banded localization seen in control conditions. In control hESCs the localization of proteins accurately mirrors patterns expected in untreated myofibrils. The reduction in contractile activity shown in cardiomyocyte sheets derived from QKI-deficient cells represents the logical outcome of the structural deficits produced in in QKI-deficient cells.

The arrangement in the Figures, pairing actin-binding probes that mark myofibers, with probes that bind Z-band proteins in the same cells emphasizes a role for QKI in disrupting both the organization of Z-bands as well as actin fibril alignment in myofibrils, illustrating that both Z-bands as well as actin fibers become disorganized. It would be helpful to indicate that troponin (TNNT2) binds to actin filaments in myofibrils and nebulin (NEBL) and alpha-actinin (ACTN2) are Z-band proteins.

The immunofluorescent imaging convincingly demonstrates the patterns of sarcomeric proteins in the parallel bundles of myofibrils formed in differentiation of control formed during hESCs to cardiomyocytes. The reduction in myofibril diameter and linear alignment evident in mutant QKI cells. The pairing of actin-binding probes with probes for Z-band proteins in Figures illustrating control, inhibited, and rescued examples of QKI effect emphasized the role of QKI in disrupting the organization of Z-bands as well as actin fiber alignment. Indicating that troponin (TNNT2) was used to identify actin filaments in myofibrils, and nebulin (NEBL) and alpha-actinin (ACTN2) used to mark Z-band proteins would emphasize how myofibril structure was altered

In conclusion, the descriptions of myofibril assembly and structure are accurately and clearly described and support a role for QKI in regulating cardiac myofibrillogenesis.

Response to reviewer 1:

To the general comments from Reviewer 1:

The authors have identified the RNA binding protein Quaking (Qki) as a key regulator of cardiomyocyte differentiation and function. Their sequencing analysis includes single-cell interrogation of cardiomyocytes differentiated from wildtype and Qki-deficient hESCs, as well as bulk RNAseq of differentiated cardiomyocytes. Their findings reveal that Quaking does not have a large impact on early cardiomyocyte specification (nor does it affect the stemness of the undifferentiated hESCs), but rather confers a significant defect in the functionality of differentiated cardiomyocytes. It is shown that differentiated cardiomyocytes which lack Qki have a suite of downregulated genes involved in sarcomere function, thus explaining the defect in differentiated cardiomyocytes. Further analysis revealed that the nuclear isoform of Qki, Qki5, is responsible for regulating a network of alternatively spliced genes which play an important role in the functionality of differentiated cardiomyocytes.

Finally, a knockout mouse model shows severe defect in myofibrillogenesis and share similar defects in splicing as the cell lines.

We thank reviewer 1 for the supportive comments and insightful suggestions. The followings are our detailed responses to the specific suggestions and comments.

To the specific comments from Reviewer 1:

1) Figure 1: The RT-PCR and WB should be performed for each QKI isoform. QKI-5, QKI-6 and QKI-7. Pan antiQKI and 'qkl' mRNA is not useful as to define which isoform is likely involved. Figure 1B, authors should add molecular weight marker. Pan-Qki antibody is used and only one band is shown, however due to the difference in sizes of the Qki isoforms -6 and -7, should we be seeing multiple bands? Western blots of specific Qki isoforms would be helpful here, even though rescue experiments with individual isoforms are provided later on.

We agree with reviewer's comment. QKI-5 is the major isoform in hESC and differentiated cardiomyocytes. As suggested, additional Western blot images was provided in the revised manuscript as supplemental information to show relative levels of QKI-5, QKI-6 and QKI-7 (Supplemental Figure 1B-E). QKI-6 and QKI-7 protein expression are significantly lower, almost at an undetectable level. Thus, although we used anti-pan-QKI antibody, Figure 1B mainly represents QKI-5 protein expression level. As shown in Supplemental Figure 1D, longer exposure of QKI-6 Western blot reveals a faint band, aligned at 37kD molecular weight marker. QKI-7 is undetectable under our Western blot condition, even at a longer exposure.

We also performed additional qRT-PCR analyses. QKI-6 and QKI-7 mRNA levels are significantly less when compared to QKI-5 mRNA level (Figure 1A), further confirming that QKI-5 is the major QKI-isoform in hESC and differentiated cardiomyocytes. This data, along with the rescue data with QKI-5, QKI-6 and QKI7 (now in Figure 8), validates that QKI-5 provides the major biological function in QKI-mediated RNA splicing in cardiomyocyte early differentiation and maturation.

2) Figure 2: The finding that H1-7 and H1-8 clones had 20% reduction in TNNT2-positive cells suggests that the differentiation defect is partially due to a loss in the number of differentiated cells. However, this does not fully explain the drastic defect in cardiomyocyte differentiation. Therefore, the scRNA analysis is useful in determining whether the defect originates in the process of differentiation. The authors argue that the scRNA analysis shows there is little difference in early lineage specification between Qkidel and WT hESCs. However, it would be much easier to visualize this claim if the authors could provide pseudotemporal trajectories of Qkidel and WT hESCs using the Monocle ordering algorithm. This would more clearly show that

the differentiation defect observed in Qkidel cardiomyocytes is not due to early specification/determination.

We appreciated the reviewer's suggestion. As suggested, we have re-characterized our scRNA-seq data with newer version of Seurat 3 and Monocle 3 algorithm. In the revision, we have provided new version of Figure 3 and have provided Monocle 3 trajectory. This new analysis, on one hand, confirmed our previous conclusion that the lineage specification and determination towards cardiomyocytes were largely preserved in hESCs-QKI_{del}; on the other hand, we found that there was a correlation that a 18.5% reduction of late cardiomyocytes (Figure 3G), apparently consistent with FACS data based on TNNT2 positivity. We also observed an increase of percentage of cardiomyocytes at transitional stage from early cardiomyocytes to late cardiomyocyte (Figure 3G). The latter data was consistent with our overall assessment that the defect was at the transition from early cardiomyocyte to late differentiated cardiomyocytes. An additional Reactome Pathway Analysis of late differentiated cardiomyocyte is also included in supplemental Figure 6B, so to further support that the altered myocardial contractile function is the major defect in QKI mutant cells.

In the original manuscript, we also provided qRT-PCR analysis on several key molecular markers for early mesodermal lineages (e.g., Brachyury T and MESP1) and cardiogenic progenitors (e.g., ISL1 and NKX2-5) at different differentiation stages (now in new supplemental Figure 6A). The fact that mesodermal and cardiomyocyte progenitor markers were normally expressed in differentiating QKI_{del} cells at Day 3 and Day 6 strongly implied that the early differentiation of hESCs-QKI_{del} towards cardiomyocyte was not affected, while the dramatic and significant downregulation of ACTN2 indicated that cardiomyocyte functional maturation was largely compromised.

3) Further, it would be expected that the differentiated cardiomyocyte population (described as being represented by clusters 0-3 at d15), should have a 20% reduction in cell number compared to the WT. Is this reflected in the scRNA analysis? This would corroborate the flow cytometry findings of reduced differentiated cell number. The scRNAseq analysis reveals no defect in the differentiation pathway, so the 20% reduction in the differentiated population remains to be addressed, are they undergoing apoptosis?

We appreciated this excellent question. Apparently, our data from new analysis is now consistent with the 20% reduction of TNNT2 positive cardiomyocytes (Figure 3E). In addition, the increased percentage of transition cardiomyocytes implies that some degree of inefficiency in the process of cardiomyocyte maturation is likely the part of the underlying mechanism.

We did not see any strong indication of apoptosis of cardiomyocytes in differentiated cells and in embryonic hearts deficient in Qki.

4) Results section "QKI-mediated alternative splicing in cardiac myogenesis": Galarneau and Richard 2005 should be referenced here when discussing the Quaking consensus sites. Supplemental figure 6 requires a bit more explanation, what target sequences were assessed for Qki binding sites? Did the authors focus only on regions which were alternatively spliced, or the full transcript? It could also be mentioned in the text for clarity that, according to the data presented in supp figure 6, the Qki core site (ACUAAAY) was enriched, while the half-site (UAAY) was not significantly enriched compared to the background sequences.

We thank the reviewer for discovering our mistake of not adequately describing the methods for previous Supplemental Figure 6 (now as supplemental Figure 7). We have added the full description of our analysis to the manuscript in Supplemental Figure 7, including the adequate references. Briefly, analysis of the half-motif was not performed because we used motifs from the CisBP-RNA binding protein motif database. Only the full motif, not the half motif, was present in the database. Therefore, it is possible that the half-motif is enriched in our alternatively spliced events, especially considering the full motif is, but in the interest of applying a standardized objective analysis pipeline we did not explore this possibility.

5) Figure 5 Panel B: Quaking binding site locations should be indicated in these splicing diagrams.

As suggested, we have indicated consensus QKI binding sites in revised Figure 6B (previous Figure 5B).

6) Panel D: RIP-qPCR shows that Qki binds to ACTN2, but an assay that would provide a more precise binding location such as ParCLIP should be done to confirm that Quaking binds to the specific Quaking binding sites which flank exon 8.

We thank the reviewer for this suggestion of using ParCLIP to map QKI binding sites. Indeed, we have made a best effort of using ParCLIP to determine QKI binding sites to several key targets in differentiated cardiomyocytes, and even tried to perform CLIP-seq analysis. However, it has been a huge challenge for us mainly due to the difficulties to obtain sufficient quantity of differentiated cardiomyocytes to perform the experiments in an affordable cost. Thus, RIP-qPCR was used as an alternative to at least to confirm that these candidates were the direct target for QKI in Figure 5C (now Figure 6C). Figure 5D (now Figure 6D) confirms that NMD mechanism is in fact the case for the downregulation of Exon 8-skipped ACTN2 mRNA.

We absolutely agree with reviewer's comment. To further address this important question, we are in the process of making mutations to these predicted QKI-binding sites by using CRISPR/Cas9 gene editing technology. Along with additional analysis of ACTN2 and other key Z-line proteins, we will be able to provide a comprehensive view of molecular regulation of sarcomerogenesis. We would appreciate very much if we are allowed to address this question thoroughly in our future work.

7) Figure 6. Panel D: The downregulation of sarcomeric proteins whose RNA levels and splicing are unaffected by Qki deficiency (presumably, an example would be MYOZ2 which is not mentioned elsewhere in the context of RNA but shows drastic reduction in protein levels), may be a symptom of the overall defective differentiated state of these cardiomyocytes, however it is also possible that Qki is regulating of RNA stability of these targets. It may be worthwhile to check the 3' UTR sequences of these sarcomeric transcripts for the presence of Qki binding sites to assess whether the functionality of Qki as a regulator of RNA stability is relevant here.

We would like to thank the reviewer for this insightful comment. MYOZ2 is known for its important function in tethering calcineurin (calcium-dependent serine-threonine phosphatase) to ACTN2 at Z-line and is known for its involvement in the pathogenesis of cardiac hypertrophy. Our RNA-seq data demonstrated that MYOZ2 has a normal RNA expression level and splicing in QKI_{del} cardiomyocytes. The dramatic downregulation in protein level is likely a secondary event to the failed myofibrillogenesis via protein degradation pathways. In addition, we did not find QKI binding site in MYOZ2.

Furthermore, it has been shown that QKI-6 and QKI-7 function are more relevant to maintaining RNA stability or other post-transcriptional activities. Inducible expression of QKI-6 and QKI-7 in QKI_{del} cells failed to rescue cardiomyocyte phenotype and MYOZ2 protein expression, which is consistent with the notion that the downregulated MYOZ2 protein is not due to defect in its RNA stability. Similarly, the broad downregulation of other sarcomere proteins, such as myosin heavy chains (MF20), cardiac troponin T (TNNT2), troponin I (TNNI3) protein levels is likely via the similar mechanism.

However, it is indeed important to determine the potential role of QKI in maintaining RNA stability. Our future analysis will be more focus on the 3' UTR-QKI binding sites in down-regulated gens.

8) Also, the WB of d15 cardiomyocytes shows that TNNT2 protein is almost completely gone in the Qki deficient cells. However, earlier flow cytometry analysis (supp fig 3) shows that there is only a 20% reduction in TNNT2-positive cells in Qki deficient d15 cardiomyocytes. Did the immunofluorescence analysis confirm the flow cytometry data by showing less TNNT2-positive cells? There is a lack of consistency here that should be addressed.

Indeed, the fluorescence signal of TNNT2 positive cells were clearly left-shifted in flow chart (now in supplemental Figure 4), indicating a reduced expression level of TNNT2 protein. We did not score the percentage of TNNT2-positive cells in our immunofluorescence staining, as the staining was used to analyze the myofibril structure. However, in general, TNNT2-fluorescence signal is dramatically less in QKI-deficient cardiomyocytes.

9) Supp Fig 10 Which antibodies are used for Qki isoforms? Please provide details in materials and methods. Qki-5, -6, and -7 are shown as all being 37kDa but should have different molecular weights, again specific labeling of molecular weight markers is important here. Western blots for Qki 6 and 7 should be included as control for Qki5 OE in Qkidel hESCs. Authors should specify what day of differentiation alternative splicing recovery (panel C) was done.

As suggested, we have added information of antibodies specific against QKI-5, QKI-6, and QKI-7 in the materials and methods.

The alternative splicing recovery was assessed at Day-15, which is now indicated in the figure legend.

Please see supplemental Figure 1 for the relative protein sizes of induced expression of QKI-5, QKI-6, and QKI-7 in hESC-QKI_{del}.

10) Figure 7 Very convincing findings that Qki5 is the isoform responsible for the phenotype. However, the finding that Qki5 induction only works at early stages of differentiation (days 0-6 but not day 8) to correct the differentiation defect contradicts earlier findings that Qki does not impact early specification of cardiomyocytes and is only involved at later, terminal stages and in the functionality of differentiated cardiomyocytes. This finding would imply that Qki does indeed play a role in early differentiation, although the effects may not be observable until terminal differentiation. This needs to be addressed.

Thanks for this excellent question. In our differentiation protocol, Day-6 is a critical transition stage that cardiogenic progenitors become towards cardiomyocyte. This transition involves a critical transcriptional switch on/off and epigenetic modulation. Simply modifying the

transition with culture condition can impact on the cardiomyocyte maturation. Dox-induction of QKI-5 expression takes at least 24 hours to reach to a level comparable to endogenous level in wild type cells. The induction of QKI-5 at Day 6 (QKI-5 reach to functional level around Day 7) perfectly distinguished the role QKI-5 in early differentiation and cardiomyocyte maturation. This data suggested that QKI-5 is not required prior to day 6-7. However, at Day 8 or 9, the QKI_{del} cells are likely genetically and epigenetically locked into a non-reversible state. Induction at Day 8 (QKI-5 expression reaches to functional level around Day 9) is likely too late to overcome the abnormal epigenetic code. We are currently in the process of verifying critical epigenetic factors contributing to this irreversible process. We will conclude this part of work as a separate manuscript in the future.

11) Supp figure 11 It should be noted that the alternative splicing of ACTN2 is not as striking in the Qkigeo/geo embryonic heart as was seen in the cell line, I would refrain from using the term "identically altered". Other targets shown earlier (fig 5a) should also be shown here to confirm similar alternative splicing patterns.

The identically altered ACTN2 alternative splicing site in hCM-QKI_{del} and QKI-deficient mouse heart was confirmed by sequencing analysis. Supp Figure 11B (now Figure 9D) was not quantitative PCR. As Exon8-skipped ACTN2 mRNA was under NMD mediated degradation in the mouse heart, it is understandable that the PCR primers amplified remaining normal form of ACTN2 much more efficiently. We provided an updated PCR data to show ACTN2 splicing pattern in supplemental Figure 9D, in which, as suggested, we also included other altered splicing genes identified in hCMs-QKI_{del}. So far, among all tested, their pattern of altered splicing events remained identical between differentiated hCMs-QKI_{del} and QKI-deficient mouse hearts.

12) The discussion of alternative splicing in the introduction is lacking appropriate references. Info required regarding the QkiBgeo mice should also appear in materials and methods.

As suggested, we have modified introduction and methods

13) This sentence is inaccurate as written. QKI does not have SH2 and SH3 domains. "QKI contains an RNA-binding motif (KH domain) and protein interaction and signaling domains (SH2 and SH3 domains), which are flanked by QUA1 and QUA2 domains."

We corrected this mistake.

14) "As we shown here, QKI is a novel RNA-binding protein that is highly expressed in developing hearts and adult hearts." Remove novel as QKI is not a novel RBP.

As suggested, we removed "novel" from the sentence.

15) The QUA domain is involved in forming homo- or heterodimers and is required for RNA binding.^{32, 33, 34, 35} Chen and Richard 1998 MCB 18:4863 should be cited instead of Darbelli and Richard 2016.

As suggested, we have modified citations.

16) QKI5 has been implicated for its major function in pre-mRNA splicing regulators,^{37, 38, 39} Delete reference 37 – they perform parCLIP to identify intronic sequences. they do not perform alternative splicing experiments. These should be cited here:

Wu et al. PNAS 99:4233
Zong et al., PloS Genet 2014 e1004289
Darbelli et al., J Neuroscience 2016
de Bruin et al., Nat Neurosci 2016

while QKI6 and QKI7 appear to be more relevant to RNA-posttranscriptional processing and transportation.³³ Primary references should be cited here.

As suggested, we have modified citations.

Response to reviewer 2:

To the general comments from Reviewer 2:

Evidence that the RNA-binding protein, QKI, plays a crucial role in myofibril formation and function is supported in convincing detail in the submitted manuscript. The **expression in vitro** of key myofibril proteins in QKI-deficient cells and in human control embryonic stem cells (hESCs), in the **cardiomyocyte differentiation** system **provided a clear view of the differences** in myofibrillogenesis **seen** in QKI-deficient cells when compared with QKI control cells. The immunofluorescent imaging shown in the Figures convincingly demonstrates that, in QKI-deficient cells, there is a notable loss of **aligned** myofibers, together with a fractured morphology of the fibers, in place of the parallel bundles of myofibrils formed during cardiomyocyte differentiation in control hESCs. The distribution of sarcomere proteins along the myofibers, additionally, is shown to be disrupted in mutant QKI cells, eliminating the characteristic banded localization seen in control conditions. In control hESCs the localization of proteins accurately mirrors patterns expected in untreated myofibrils. The reduction in contractile activity shown in cardiomyocyte sheets derived from QKI-deficient cells represents the logical outcome of the structural deficits produced in in QKI-deficient cells.

The arrangement in the Figures, pairing actin-binding probes that mark myofibers, with probes that bind Z-band proteins in the same cells emphasizes a role for QKI in disrupting both the organization of Z-bands as well as actin fibril alignment in myofibrils, illustrating that both Z-bands as well as actin fibers become disorganized. It would be helpful to indicate that troponin (TNNT2) binds to actin filaments in myofibrils and nebulin (NEBL) and alpha-actinin (ACTN2) are Z-band proteins.

The immunofluorescent imaging convincingly demonstrates the patterns of sarcomeric proteins in the parallel bundles of myofibrils formed in differentiation of control formed during hESCs to cardiomyocytes. The reduction in myofibril diameter and linear alignment evident in mutant QKI cells. The pairing of actin-binding probes with probes for Z-band proteins in Figures illustrating control, inhibited, and rescued examples of QKI effect emphasized the role of QKI in disrupting the organization of Z-bands as well as actin fiber alignment. Indicating that troponin (TNNT2) was used to identify actin filaments in myofibrils, and nebulin (NEBL) and alpha-actinin (ACTN2) used to mark Z-band proteins would emphasize how myofibril structure was altered

In conclusion, the descriptions of myofibril assembly and structure are accurately and clearly described and support a role for QKI in regulating cardiac myofibrillogenesis.

We thank reviewer 2 for the supportive comments. As suggested, we indicated that TNNT2 is a part of myofibril structure that binds to Z-band proteins in the text.

Response to reviewer 3:

To the general comments from Reviewer 3:

This is an interesting study about the role of a RNA binding protein QKI in cardiac development. I will focus my review on the aspects related to scRNA-seq.

This is a very nice scRNA-seq analysis and the authors have made it clear about what they have done in the Methods section. Kudos to them.

We thank reviewer 3 for the supportive comments. The followings are our detailed responses to the specific suggestions and comments.

To the specific comments from Reviewer 3:

1) I am curious by the many cardiac subpopulations the authors have found in their cultures: 8 cardiac clusters in day 6 cells and 4 clusters in day 15 cells. Do they represent different stages of cardiac differentiation? (Day 6 clusters 0, 4 and 7 appear to express less TNNT2 than the other cardiac clusters – are they more toward the progenitor stage?) Or are they related to different lineages like first vs. second heart fields, atrial vs. ventricular? I am interested to see the differences among these clusters. A cell fate trajectory analysis like Monocle might also be helpful.

Yes, indeed, they are presented at different stages in the process of differentiation, from early progenitors to cardiogenic progenitors and to differentiated cardiomyocyte. As suggested, a new Monocle 3 trajectory analysis is included in the revised manuscript (Figure 3).

We particularly checked if we can identify specific clusters for ventricular or atrial cardiomyocytes. However, as our analysis was ended at Day-15, these cardiomyocytes in different clusters express both atrial and ventricle genes, suggesting that our unsupervised clustering reflected more of functional states of the cells at this stage, instead of anatomical location.

Cluster 3 cells (Day-6) in original data have some unique signatures of first heart field cardiomyocytes. In our study, as we did not observe any major difference between normal control and QKI-deficient differentiated cells at Day 6, we believed QKI did not play any significant role in early specification of cardiomyocyte in first and second heart fields.

2) QKI KO cells may adopt alternative cell fates, as suggested by a reduction in cTnT+ cells by day 15. Therefore, I expect to see some clusters that primarily consist of KO cells. However, it appears to be the other way around: cluster 8 in day 6 and cluster 6 in day 15 are mostly from wildtype cells. They don't seem to be cardiac. What are they?

Previous cluster 8 (Day-6) and cluster 6 (Day 15) were small clusters, representing cells towards neuronal lineage (now as presented as cluster 17). Interestingly, data from our recent on-going experiment using neuronal differentiation protocol suggesting a strong defect in neural progenitor differentiation in hESC-QKI^{del}, which is surprisingly consistent.

(Note: although the differentiation protocol is to promote cardiomyocyte differentiation, there is always a small portion of cells commits to other cell types.)

3) The authors used 40 μ m strainers for their cardiomyocyte prep. I assume the wildtype cardiomyocytes are more mature and perhaps bigger since they beat better. Will the 40 μ m strainer be more likely to filter out the more matured cardiomyocytes? If so, there might be even a bigger difference between wildtype and KO cardiomyocytes.

This is not the major concern in our case. The 40 μ m strainer is sufficient for almost all hESC-derived cardiomyocytes isolated from mono-layer sheet, as the cells are usually no bigger than 20 μ m in size in cell suspensions at Days-6 and -15.

REVIEWERS' COMMENTS

Reviewer #1 (Remarks to the Author):

The authors have adequately addressed the suggested revisions and I agree with their rebuttal. Accept.

Some revisions were not addressed, such as assessing the presence of the Qki half site (not just the core site), defining the exact binding sequencing of Qki in the sarcomeric target proteins with PARclip, and investigating the effect of Qki isoform-specific deletion on mRNA stability of the sarcomeric targets. These are important molecular questions that remain to be addressed in future studies, however they are not essential to the significance of these findings.

Reviewer #2 (Remarks to the Author):

NCOMMS-19-41651A

I have received the manuscript and Figures for "RNA-binding protein QKI is a critical pre-mRNA alternative splicing regulator of cardiac myofibrillogenesis and contractile function". I am satisfied that the section I reviewed is accurate and convincing in describing the Z-bands and actin fibers of the QKI deficient cells and the control cells.

Reviewer #3 (Remarks to the Author):

This reviewer appreciates the additional efforts of the authors to provide a Monocle analysis of the single-cell RNA-seq data. It is now clearer to see the differentiation defects in KO cells. However, I have trouble understanding how the 6 clusters (7 if you include the non-CM one) in Monocle are defined. Are they defined by some markers? If so, I am unable to find them.